# Evaluation of Next-Generation Sequencing Applied to *Cryptosporidium parvum* and *Cryptosporidium hominis* Epidemiological Study

**DOI:** 10.3390/pathogens11080938

**Published:** 2022-08-18

**Authors:** Eloïse Bailly, Stéphane Valot, Anne Vincent, Yannis Duffourd, Nadège Grangier, Martin Chevarin, Damien Costa, Romy Razakandrainibe, Loïc Favennec, Louise Basmaciyan, Frédéric Dalle

**Affiliations:** 1Parasitology-Mycology Laboratory, University Hospital Biology Platform, Dijon University Hospital Center, 21000 Dijon, France; 2Associated Laboratory CNR-LE for Cryptosporidiosis, University Hospital Biology Platform, Dijon University Hospital Center, 21000 Dijon, France; 3Inserm UMR 1231 GAD, Genetics of Developmental Disorders, Bourgogne-Franche Comté University, FHU TRANSLAD, 21000 Dijon, France; 4CNR LE for Cryptosporidiosis (National Center of Reference for Cryptosporidiosis, Expert Laboratory), Santé Publique France, Rouen University Hospital Center, 76031 Rouen, France; 5Laboratoire de Parasitologie-Mycologie, Centre Hospitalo-Universitaire de Rouen, 76031 Rouen, France; 6EA ESCAPE 7510, Rouen University of Medicine Pharmacy, 76031 Rouen, France; 7UMR PAM, Bourgogne Franche-Comté University—AgroSup Dijon—Team VAlMiS (Food, Wine, Microbiology and Stress), 21000 Dijon, France

**Keywords:** Next-Generation Sequencing, *Cryptosporidium* sp., cryptosporidiosis, epidemiology, subtyping, genetic diversity

## Abstract

**Background**. Nowadays, most of the *C. parvum* and *C. hominis* epidemiological studies are based on gp60 gene subtyping using the Sanger sequencing (SgS) method. Unfortunately, SgS presents the limitation of being unable to detect mixed infections. Next-Generation Sequencing (NGS) seems to be an interesting solution to overcome SgS limits. Thus, the aim of our study was to (i) evaluate the reliability of NGS as a molecular typing tool for cryptosporidiosis, (ii) investigate the genetic diversity of the parasite and the frequency of mixed infections, (iii) assess NGS usefulness in *Cryptosporidium* sp. outbreak investigations, and (iv) assess an interpretation threshold of sequencing data. **Methods**. 108 DNA extracts from positive samples were sequenced by NGS. Among them, two samples were used to validate the reliability of the subtyping obtained by NGS and its capacity to detect DNA mixtures. In parallel, 106 samples from French outbreaks were used to expose NGS to epidemic samples. **Results**. NGS proved suitable for *Cryptosporidium* sp. subtyping at the gp60 gene locus, bringing more genetic information compared to SgS, especially by working on many samples simultaneously and detecting more diversity. **Conclusions**. This study confirms the usefulness of NGS applied to *C. hominis* and *C. parvum* epidemiological studies, especially aimed at detecting minority variants.

## 1. Introduction

*Cryptosporidium* sp. are ubiquitous parasites that infect a broad range of hosts including humans and animals [1]. This protozoan parasite’s transmission can be zoonotic or anthroponotic (i.e., human to human, animal to human, and animal to animal contact), occurring after *Cryptosporidium* sp. oocysts ingestion (i) through contaminated aliments or water or (ii) by a direct feco-oral route [1]. *Cryptosporidium* sp. is responsible for enteric diseases that may be severe, chronic, and life-threatening in immunocompromised humans [1,2]. Resistance in the environment, low infective doses, as well as the ability of oocysts to withstand chlorination, enhance the spread of this parasite and thus the emergence of sporadic and epidemic cryptosporidiosis cases [3,4,5,6,7,8,9]. In France, large water- and food-borne outbreaks have been reported in the last few years, highlighting the widespread distribution of the parasite and its important health impact [5,6,7,8,9]. More recently, in 2019, a *Cryptosporidium* sp. outbreak occurred in Grasse, south of France, after the contamination of drinking water supplies, affecting almost 20,000 people and depriving them of drinking tap water for several months [10]. Overall, epidemiological investigation reports worldwide have highlighted the need to improve the diagnosis, epidemiological knowledge, and risk estimation of cryptosporidiosis. In this context, a French surveillance network has been progressively set up, leading to the creation of the National Reference Center-Expert Laboratory (CNR-LE) in 2017 for cryptosporidiosis, whose missions include diagnosis expertise, epidemiological surveillance, infection control, and epidemic alert [5]. Thus, rapid and reliable detection and characterization methods are needed for the CNR-LE for cryptosporidiosis to fulfill its missions. In recent years, numerous molecular biological tools have been developed to detect and differentiate *Cryptosporidium* species, improving epidemiological knowledge of cryptosporidiosis [1,11,12]. These tools include PCR-based genotyping, Sanger sequencing of PCR products, restriction fragment length polymorphism (RFLP) analysis of PCR products, and qPCR assays using fluorescent probes and melting curves analysis [12]. Thus, the development of molecular techniques has notably allowed researchers to show that, while *C. parvum* and *C. hominis* are reported as being responsible for 96% of the cases of cryptosporidiosis in France and worldwide, at least 20 other *Cryptosporidium* species are able to infect humans [5,11]. For instance, in France, the seven most frequently isolated species between 2017 and 2019 include *C. parvum* (72%), *C. hominis* (24%), *C. felis* (2%), *C. cuniculus* (>1%), *C. meleagridis* (<1%), *C. canis* (<1%), and *C. ubiquitum* (<1%) [5]. In addition, the development of subtyping molecular methods has offered the possibility to further characterize *C. parvum* and *C. hominis* within-species genetic diversity [3,11]. Subtype families have been described, differing from each other by the gene sequence of the highly polymorphic gene coding for a 60 kDa glycoprotein (gp60) [11,13]. Furthermore, the gp60 gene has a microsatellite sequence at the 5′ end, consisting of tandem repeats of the serine-coding trinucleotide TCA, TCG, and TCT, which varies within subtype families depending on the number of trinucleotide repetitions [11,13]. Such subtyping classification of *C. hominis* and *C. parvum* isolates has proved helpful (i) in epidemiological and environmental reports, highlighting geographical and host distribution specificities of *Cryptosporidium* isolates [12], as well as virulence features associated with specific subtypes [14], and (ii) in outbreak investigation and management aimed at specifying the origin of contamination, relating transmission vehicles, and linking cases [3]. Until recently, most gp60 genotyping-based epidemiological studies relied on qualitative PCR followed by Sanger-based sequencing methods and further automated fragment analysis to specify the subtype of *C. hominis* and *C. parvum* [11,12]. Unfortunately, Sanger sequencing (SgS) methods have proved unsuitable to detect mixtures of subtypes without additional molecular cloning methods, leading to possible underestimation of the prevalence of *Cryptosporidium* species/subtypes in mixed infections [15,16]. Alternative approaches have been proposed using deep sequencing amplicon-based technologies, i.e., Next-Generation Sequencing (NGS). These sequencing approaches have already demonstrated their added value in all fields of microbiology, notably for the sequencing and detection of viral genomes, including SARS-CoV-2 [17]. Contrarily to SgS methods, the NGS-based gp60 genotyping method at adequate sequencing depths has proved successful in resolving mixtures of amplicons, allowing *C. hominis* and *C. parvum* subtyping in some studies [15,16,18,19,20,21]. Each of these studies worked on various matrices, with various goals, which shows the wide possibilities offered by the NGS. However, these approach disparities also show a current lack of standardization regarding NGS applied to *Cryptosporidium* sp. subtyping, notably concerning a threshold to interpret the sequencing data. In this context, the aim of our study was to (i) assess the reliability of the NGS as a molecular typing tool for *C. parvum* and *C. hominis*, (ii) evaluate its added value in the study of genetic diversity and frequency of mixed *C. parvum* and *C. hominis* infections, (iii) assess its usefulness in outbreak investigations, and (iv) assess an acceptable interpretation threshold for sequencing data. 

## 2. Results

### 2.1. NGS Protocol Gives Reliable Subtyping Results and Allows Detecting DNA Mixtures of Cryptosporidium *sp.*

The 16S Metagenomics Sequencing Library Preparation (16SMSLP) protocol (Illumina, San Diego, CA, USA) has been used in several studies applying NGS for *Cryptosporidium* sp. subtyping. However, this protocol was used with various bioinformatics pipelines and databases. Thus, we wanted to confirm that the 16SMLP NGS protocol suits *Cryptosporidium* sp. subtyping when associated with the DADA2 pipeline and our in-house database, by studying unmixed DNA samples (1572 and 2055) and comparing NGS data with subtyping results formerly obtained at the gp60 gene locus using the SgS method. The sequencing of sample 1572 (i.e., *C. parvum*-IIcA5G3) alone generated 241,302 quality sequences, with 240,932 of them corresponding to *C. parvum*-IIcA5G3 and 93 sequences corresponding to C. hominis-IbA10G2. In parallel, the sequencing of sample 2055 (i.e., *C. hominis*-IbA10G2) generated 200,284 quality sequences, with 200,160 corresponding to *C. hominis*-IbA10G2 and 17 sequences corresponding to *C. parvum*-IIcA5G3. All in all, NGS found the same results for major subtypes previously identified by SgS for both samples, showing that the chosen NGS protocol was suitable for subtyping, but also found unexpected minor subtypes. Indeed, 116 and 59 sequences attributed to *C. parvum*-IIaA18G1R1 and *C. parvum*-IIdA22G1 were obtained in samples 1572 and 2055, respectively. 

Regarding the ability of NGS to detect DNA mixtures of *Cryptosporidium* sp., using mixes A to G (Supplementary Material Appendix A), NGS sequencing generated sequences attributed to *C. parvum*-IIcA5G3 and *C. hominis*-IbA10G2 for all mixtures, including those containing a minority subtype with the lowest proportion (i.e., mix D and mix G). As in 1572 and 2055 samples, additional *C. parvum* subtypes, i.e., *C. parvum*-IIaA18G1R1 and *C. parvum*-IIdA22G1, were identified in all of the mixtures, including the negative control. It is of importance to note that all of these sequences corresponded to *C. parvum* subtypes that were handled in the same run. On the basis of these observations, highlighting highly probable DNA cross-contamination between samples, the presence of the negative control proved to be essential for the interpretation of the data. The sequencing of negative controls revealed the presence of 1210 sequences (i.e., the average value of the three controls) of which 225 matched *C. parvum*-IIcA5G3 and *C. hominis*-IbA10G2. The most represented subtype in the negative control was *C. parvum* IIdA22G1 reaching 983 sequences. Two hypotheses could explain the high number of these subtype sequences detected in the negative control: (i) Almost half of the samples handled in the run were positive for this subtype, and (ii) as previously described, the prevalence of contaminants can be higher in negative controls than in other samples due to the absence of competitive DNA during the sequencing process [22]. This highest value observed in the negative control was thus used as the interpretation threshold. Thereby, any sequence generated by NGS with 983 sequences or less was considered a contaminant. Note, however, that while mixes C and D were known to contain 1% and 0.1% of *C. hominis* IbA10G2, respectively, only 777 and 75 sequences, respectively, were generated by NGS. Some sequences generated during the NGS run could not be assigned to any sequence of our database, nor to BLAST. No sequences other than *Cryptosporidium* sp. were identified, confirming the absence of environmental contaminations. All the results are summarized in Table 1. 

### 2.2. Application of NGS for Cryptosporidium *sp*. Subtyping Outbreaks Isolates

We then applied the NGS protocol to four samples formerly characterized during an outbreak that occurred in Divonne-les-Bains, France [23] (Table 2). The initial investigation in 2003 identified mixed infection involving *C. parvum* and *C. hominis* thanks to a molecular cloning step followed by SgS of the clones, as previously described [7]. Note that the gp60 subtypes had not been carried out during the first epidemic investigation [23]. As observed during the validation step of the protocol, unexpected sequences corresponding to other samples manipulated in the same run were found in low proportions in each sample. Thus, the sequencing data were interpreted using the highest value (983 sequences) observed in the negative control. Sequencing of the samples by NGS showed similar results as the initial investigation for samples D24 and D26. Surprisingly, while a mixture of *C. parvum* and *C. hominis* was expected in sample D45, only 84 sequences of *C. hominis* were generated, compared to the 983-sequence threshold set relative to the negative control. Finally, only a small number of sequences were generated for sample D23, making the subtyping uninterpretable. This could be explained by the fact that D23 was weakly positive in Real-Time PCR at the 18S rRNA locus when the integrity of DNA was verified at the beginning of our work. 

Then, 102 DNA extracts from Grasse outbreak in 2019 [10] were sequenced by NGS at the gp60 gene locus. All samples generated quality sequences (Appendix A). The predominant subtype identified was *C. parvum*-IidA22G1 (n = 91), corresponding to the one identified by SgS on 48 samples during the initial investigation. Other variants were identified in 11 samples, in concordance with those obtained with the initial SgS. Note that one sample (2249) revealed a DNA mixture with 51,350 sequences of *C. parvum*-IIdA22G1, 332,030 sequences of *C. parvum*-IIaA15G2R1, and a minor subtype of *C. parvum*-IIaA11G2R1 (Appendix A). Again, additional sequences in lower or equivalent proportions than in the negative control were observed, corresponding to other *C. parvum* subtypes handled in the same run. Some sequences generated during the NGS run could not be assigned to any sequence of our database, nor to BLAST. Finally, for six samples, a predominant subtype was associated with an unexpected minority subtype, *C. parvum*-IIaA11G2R1, with an average of 1700 sequences per sample, which was not observed by the former SgS (Appendix A). Figure 1 compares the investigation of the outbreak carried out by SgS and NGS, in terms of subtype relative proportions and mixed-infection detection. By investigating more samples, NGS made it possible to reflect the true proportions of each variant within the outbreak, notably by showing that *C. parvum* IIdA22G1 represented almost 90% of the subtypes found during the outbreak. Furthermore, NGS was able to highlight mixed infections within the outbreak, which had been ignored by SgS. 

## 3. Discussion

Subtyping is an interesting tool for understanding outbreak dynamics, contamination sources, geographic distribution, etc. [3,11,12]. In the past few years, many epidemiological studies have focused on Sanger sequencing methods in order to study *Cryptosporidium* sp. subtypes present in different matrices. However, Sanger methods have some limitations, in particular the inability to overcome complex DNA mixtures and identify variants present in low proportions [15,16]. Recent studies have reported NGS as a suitable tool for subtyping *C. parvum* and *C. hominis* isolates while having the ability (i) to detect complex DNA mixtures of *Cryptosporidium* sp. and (ii) to investigate large numbers of samples simultaneously [15,16,18,19,20,21,24]. Other techniques are able to highlight *Cryptosporidium* species/subtypes mixed infections; this is the case for capillary electrophoresis (CE)-based DNA fragment analysis tool. This technique, described as time- and cost-effective, is, however, limited by the fact that fragments from different species/subtypes but of equal length cannot be differentiated [25]. Thus, NGS methods represent a potential molecular tool for the investigation of *Cryptosporidium* sp. outbreaks and the study of *Cryptosporidium* sp. genetic diversity. The current lack of standardization first led us to validate the chosen NGS protocol, in order to confirm its reliability and study its performance. We then confronted NGS with clinical samples to assess its potential benefit in the study of *Cryptosporidium* sp., and to try to set an interpretation threshold. 

NGS gave reliable results for *C. parvum* and *C. hominis* gp60 subtyping by using the association of the 16SMSLP protocol with our pipeline and database, and found results identical to previous SgS results obtained in two samples. Then, as no study has validated the performances of the 16SMSLP protocol with calibrated mixtures to our knowledge, we used in vitro *C. parvum*-IIcA5G3 and *C. hominis*-IbA10G2 mixtures, as well as clinical samples displaying DNA mixtures of *C. parvum* and *C. hominis*. This confirmed the NGS capacity and that the SgS method is not the best-suited technique to achieve genetic diversity. Note, however, that previous studies have shown that the preferential amplification of *C. parvum* DNA compared to *C. hominis* DNA exists by PCR methods [26]. This certainly explains that, in equal proportions, *C. parvum* generated more sequences by NGS than *C. hominis* in the mixtures A to G.

Aimed at fully exploiting the ability of NGS to deal with large numbers of samples simultaneously, 102 clinical samples isolated during the same outbreak were finally tested. During the initial epidemiological investigation in Grasse, subtyping was conducted on only 48 samples because of the time-consuming nature of the SgS, whereas only two runs of NGS were sufficient to investigate the entire outbreak, highlighting NGS as an efficient tool to quickly explore cryptosporidiosis outbreaks. 

Moreover, NGS sequencing reported additional variants compared to the SgS. Indeed, one sample showed a mixture of three variants of *C. parvum*, two in high proportions (i.e., IIdA22G1 and IiaA15G2R1) and one in a low proportion (i.e., IIaA11G2R1) when only one subtype had initially been identified by the initial SgS. Surprisingly, five other samples showed this identical minority subtype (i.e., IIaA11G2R1) at an average of 1700 sequences per sample. As this subtype was not detected in the negative control nor in any other sample tested in the same run, this excludes the possibility of cross-contamination during library preparation. Interestingly, those samples were isolated at the end of the outbreak, which suggests the end of the epidemic wave, highlighted by the appearance of greater diversity in *Cryptosporidium* isolates. 

However, limitations regarding the interpretation of the data were observed using NGS methods. Indeed, the presence of unexpected sequences in low numbers in all the samples, including negative controls, highlighted the need for operator expertise and technical care since most of those unexpected sequences were linked to *Cryptosporidium* species/subtypes present in other samples handled in the same run. The negative control thus appeared essential for the interpretation of the data, by setting an interpretation threshold based on the highest number of sequences generated in the negative control during each sequencing run. Indeed, this is not acceptable to consider all the sequences generated during a sequencing run, with the risk of identifying subtypes as true when they are not. The use of the negative control to set a detection threshold is, however, questionable since cross-contaminations can lead to misinterpretations. For example, for sample D45 and mixes C and D, the detection of *C. hominis* IbA10G2 sequences was expected but sequencing generated fewer sequences than in the negative control. These observations indicate that minority variants present at 1% for *C. hominis* would have been possibly ignored when setting up an interpretation threshold as described in our study. In other words, such a threshold allows an increase in specificity but causes a loss in detection sensitivity. Finally, these observations highlight the difficulty of setting up a threshold for the interpretation of NGS data and the need for standardization of interpretation thresholds to allow for the comparison of the results of genotyping studies carried out with the same NGS-based tools. Paparini et al. used negative controls displaying three sequences when most of their samples displayed mixtures of several variants including one dominant variant and rare variants (i.e., ≤9 sequences) [16]. The authors decided to set a threshold of nine sequences, which was higher than the maximum number of sequences observed in the negative controls (i.e., three). In another study, the authors set an arbitrary threshold to 0.01% of the total number of sequences, under which sequences were considered non-specific [15]. Applying such a threshold to our study would have considered *C. parvum*-IIdA22G1 as true subtypes in mixes A to G when their presence in our negative controls prompted us to consider these sequences of cross-contamination origin. DeMone et al. in 2020 applied NGS for the detection of protozoan pathogens in shellfish. They suggested that the cut-off should be a multiplier (10× or 100×) of background protozoa read levels detected in the negative controls [27]. A higher cutoff would minimize false positives; however, some true positives could be missed. A lower cutoff may increase the number of false positives but will minimize false negatives. If we chose a threshold at 10× the maximum number of reads obtained in our negative controls (i.e., 983 sequences), we would consider the sequences attributed to *C. parvum*-IidA22G1 and -IiaA18G1 as contamination during the sequencing of samples 1572, 2055, and DNA mixtures, which suits our interpretation. However, we should consider that mixes C, D, F, and G do not contain *C. hominis*-IbA10G2 or *C. parvum-*IicA5G3, whereas we know that they do (0.1–1%). This would therefore mean that we accept ignoring minority variants present at 1%. Overall, these observations confirm the difficulty of interpreting sequences generated in low proportions and the need for additional assays and standardized practices. 

In parallel, cross-contamination could be excluded for some unexpected sequences thanks to original features shared by these subtype sequences (i.e., IiaA11G2R1). The presence of minor subtypes that cannot be linked to cross-contamination (such as *C. parvum* IIaA11G2R1) should raise questions about sequencing errors. Sequencing errors could be linked to (i) PCR polymerase slippage as suggested elsewhere [24], linked to polymerase elongation mistakes in microsatellites regions, or (ii) sequencing errors linked to the MiSeq sequencer, which is known to have an error rate of less than 1% [28]. However, that random sequencing errors could generate an identical subtype in six samples seems highly unlikely. Additionally, the DADA2 bioinformatics pipeline is supposed to limit the impact of PCR/sequencing errors on the final data. This minority variant could therefore reflect the end of the clonal epidemic wave, or could be linked to within-host genetic diversity, resulting from mutations or recombination during the parasitic life cycle [15].

In conclusion, this study confirmed that NGS answers the current need for *Cryptosporidium* sp. subtyping by targeting the gp60 gene. It provides useful information, particularly with regard to DNA mixtures, allowing the handling of a large number of samples simultaneously, which is crucial for outbreak investigations. The use of a negative control in each run appears essential to interpreting sequencing data. However, interpretation difficulties persist for low-proportion variants, pointing to the need for standardization. In addition, note that new primers have recently been designed, aimed at targeting other *Cryptosporidium* species potentially able to infect humans, that are not targeted by primers and tools presented in this study [29,30].

## 4. Materials and Methods

### 4.1. Sample Collection

A total of 108 DNA samples extracted from positive stools and collected from 2003 to 2019 were retrospectively included in this study (Appendix A). Those samples had initially been characterized as part of the National Reference Center–Expert Laboratory (CNR-LE) for the cryptosporidiosis framework. Of the 108 samples, 2 samples (i.e., isolates 1572 and 2055) were used (i) to validate the reliability of our chosen NGS protocol (i.e., the association of library preparation protocol, bioinformatics pipeline, and database) and (ii) to study the ability of NGS to detect *C. parvum* and *C. hominis* DNA mixtures. The other 106 samples, obtained from two distinct French outbreaks, were studied to assess the usefulness of NGS as a tool for outbreak investigations: 4 DNA extracts from an outbreak that occurred in Divonne-les-Bains, in 2003 [23], and 102 DNA extracts from an outbreak that occurred in Grasse, in 2019 [10] (Figure 2). To optimize the use of NGS flowcells, samples 1572 and 2055 and mixes A to G were handled in the same NGS run as Divonne les Bains samples. All in all, results obtained with the 108 samples were used to assess an acceptable interpretation threshold. DNA samples were conserved at −80 °C since the initial diagnostic. 

### 4.2. Next-Generation Sequencing (NGS) Protocol 

The NGS sequencing at the gp60 gene locus was performed using the “16S Metagenomics Sequencing Library Preparation” (16SMSLP) protocol following recommendations by Illumina^®^ (Illumina, San Diego, CA, USA), and using two-step PCR amplification (PCR A and PCR B) of the gp60 locus to gain sensitivity (Appendix A). Briefly, primers for PCR A were AL3532 (5′-TCCGCTGTATTCTCAGCC-3′) and LX0029 (5′-CGAACCACATTACAAATGAAGT-3′) [13]. Primers for PCR B were modified by MiSeq adapter sequences on the 5′ end as described by the manufacturer and were as follows: 5′ TCGTCGGCAGCGTCAGATGTGTATAAGAGACAG–{AL3532} and 5′ GTCTCGTGGGCTCGGAGATGTGTATAAGAGACAG–{LX0029}. Each reaction (25 µL) contained 2.5 µL of DNA (DNA extract from stool for PCR A, amplified DNA for PCR B), 12.5 µL of the HotStarTaq master mix Qiagen (containing 1× PCR buffer, 2.5U DNA polymerase, 200 µM of each dNTP, and 1.5 mM MgCl2), 10 µM of each primer, and 9 µL of nuclease-free water. The PCR cycling conditions were almost the same for PCRs A and B, consisting of an initial denaturation step at 95 °C for 15 min, then 45 cycles and 35 cycles at 94 °C for PCR A and B, respectively, 50 °C for 45 s, 72 °C for 59 s, and 72° for 10 min. PCR A and B were performed in an ABI Applied Biosystems™ 9700 thermocycler. Amplicons of PCRs A and B were verified by a 2% agarose gel migration. Sequencing libraries were then prepared as follows: PCR amplicons from PCR B were purified twice using the Agencourt AMPure XP Bead PCR purification protocol (Beckman Coulter Genomics), with an intermediate indexation PCR, then pooled in approximate equimolar ratios. Sequencing was performed on an Illumina MiSeq using (i) a 600-cycle (300 paired-end reads) V3 reagent kit following the manufacturer’s recommendations and (ii) PhiX as a control (Illumina, San Diego, CA, USA) included in each run. Three no-template PCR controls, consisting of DNA-free molecular grade water, were included during the library preparation and distributed between samples in the PCR plate layout. Two DNA samples (1572 and 2055) with low and similar Ct values, of distinct species/subtypes (i.e., *C. parvum*-IIcA5G3 and *C. hominis*-IbA10G2), were mixed as described in Appendix A in order to investigate the performances of the 16SMSLP protocol to detect calibrated mixtures of *Cryptosporidium* sp, while all the other samples were sequenced pure. The integrity of DNA was confirmed using Real-Time PCR at the 18S locus [31]. 

### 4.3. Bioinformatics Analysis

Bioinformatics analysis of amplicon sequences was performed using the DADA2 pipeline [31]. This pipeline is supposed to limit the impact of PCR/sequencing errors on the final data. DADA2 is described as a “correcting Illumina-sequenced amplicon errors” pipeline, which proceeds to the modeling of the sequencing error, and is supposed to make it possible to distinguish mutant sequences from erroneous sequences. Thus, as described by Callahan et al. in several mock communities, DADA2 identified more real variants and output fewer spurious sequences than other methods [32]. The starting point of the pipeline was a set of Illumina-sequenced paired-end fast files that were split by sample and from which the barcodes/adapters are removed. The end product was an amplicon sequence variant (ASV) table, which records the number of times each exact amplicon sequence variant was observed in each sample. Next, the pipeline inspected the read quality profiles and error rates at each base position, which allowed us to trim the reads where the quality distribution is below a given threshold (Phred score < 30). Then, forward and reverse reads were merged together to obtain the full denoised sequences (the overlap region must be at least 12 identical bases). Chimeras were removed, and taxonomy was assigned to the sequence variants, using the naive Bayesian classifier method. Quality sequences obtained by NGS for the 108 samples were compared with an in-house reference database. This database was obtained using the one described by Zahedi et al., which was initially made of 131 sequences of *Cryptosporidium* sp. subtypes [15]. Sequences were retrieved in GenBank, and the database was completed with supplemental subtypes characterized in the Parasitology Laboratory of the University Hospital Center of Dijon. The final database was composed of 156 reference nucleotidic sequences of *Cryptosporidium* sp. (Appendix A), gathered in a FASTA format file and implemented in the pipeline. Sequences that were not assigned to the reference database were compared to the “Nucleotide Database Collection” using the BLAST algorithm to obtain identification [33].

## Figures and Tables

**Figure 1 pathogens-11-00938-f001:**
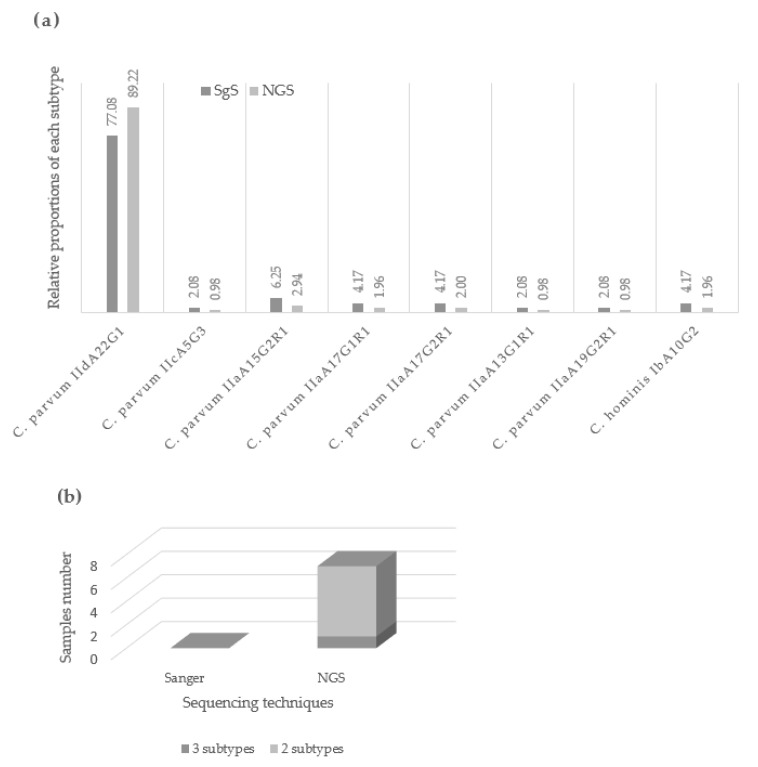
Investigation of Grasse outbreak by SgS and NGS. (**a**) Relative proportions attributed to each subtype of Grasse outbreak depending on the investigation technique. Due to the time-consuming nature of SgS, only 48 samples were sequenced by this technique during the initial investigation of the outbreak. Whereas NGS, by working on a large number of samples at the same time, allowed us to sequence all the samples of the outbreak in only two runs. By investigating more samples, NGS made it possible to reflect the true proportions of each variant within the outbreak, which is important for the epidemiological understanding of the parasite. (**b**) Mixed infections detected during Grasse outbreak depending on the investigation technique. Among the 102 samples from the Grasse outbreak, NGS revealed six mixed infections, including one involving three subtypes. These DNA mixtures were not detected by SgS.

**Figure 2 pathogens-11-00938-f002:**
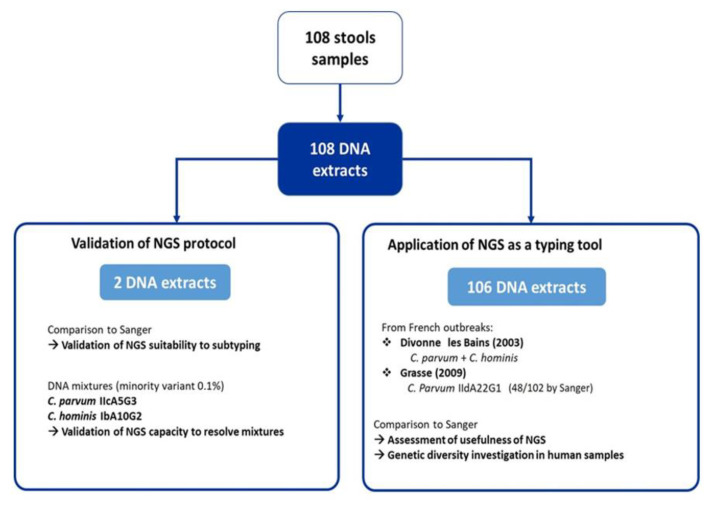
Flowchart of the study.

**Table 1 pathogens-11-00938-t001:** Results of Next-Generation sequencing for *C. parvum* IIcA5G3/*C. hominis* IbA10G2 DNA mixtures.

		*C. parvum*_IIcA5G3		*C. hominis*_IbA10G2		*C. parvum*_IIaA18G1R1		*C. parvum_*IIdA22G1		Total (Not Assigned)
Samples	Expected Proportion of *C. parvum*_IIcA5G3/*C. hominis*_IbA10G2	Number of Sequences	%	Number of Sequences	%	Number of Sequences	%	Number of Sequences	%	
**1572**	100%/0%	240,932	99.85	93	0.04	19	0.01	97	0.04	241,302 (161)
**2055**	0%/100%	17	0.01	200,160	99.94	13	0.01	46	0.02	200,284 (48)
**mix A**	50%/50%	117,150	74.07	40,930	25.89	0	0	33	0.02	158,113 (0)
**mix B**	90%/10%	178,170	94.61	10,040	5.33	27	0.01	49	0.03	188,317 (31)
**mix C**	99%/1%	155,550	99.44	777	0.50	23	0.01	63	0.04	156,431 (18)
**mix D**	99.9%/0.1%	163,383	99.91	75	0.05	22	0.01	27	0.02	163,534 (27)
**mix E**	10%/90%	81,076	31.19	178,697	68.75	0	0	108	0.04	259,924 (43)
**mix F**	1%/99%	6280	2.31	265,724	97.63	33	0.01	105	0.04	272,165 (23)
**mix G**	0.1%/99.9%	1140	0.46	247,914	99.48	0	0	133	0.05	249,215 (28)
**CTRL neg**	0%/0%	94	7.76	131	10.82	2	0.16	983	81.2	1210 (0)

**Table 2 pathogens-11-00938-t002:** Results of the NGS sequencing of the four samples obtained from Divonne les Bains outbreak.

		*C. parvum*_IIcA5G3		*C. hominis_*IbA10G2		*C. parvum*_IiaA18G1R1		*C. parvum*_IidA22G1		Total (Not Assigned)
Samples	Expected Results	Number of Sequences	%	Number of Sequences	%	Number of Sequences	%	Number of Sequences	%	
**D23**	*C. hominis* + *C. parvum*	127	48.46	58	22.14	44	16.79	33	12.6	262 (0)
**D24**	*C. hominis*	82	0.04	206,058	99.88	59	0.03	43	0.02	206,309 (67)
**D26**	*C. hominis* + *C. parvum*	60,215	63.04	35,172	36.82	31	0.03	16	0.02	95,512 (78)
**D45**	*C. hominis* + *C. parvum*	296,688	99.91	84	0.03	0	0	101	0.03	296,954 (81)
**CTRL neg**	Negative	94	7.76	131	10.82	2	0.16	983	81.2	1210 (0)

## Data Availability

Not applicable.

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
