# Peer review of "Evaluation of Next-Generation Sequencing Applied to Cryptosporidium parvum and Cryptosporidium hominis Epidemiological Study"

_pathogens, 2022, doi:10.3390/pathogens11080938_

Round 1

Reviewer 1 Report

The manuscript (ID-1787620, pathogens) entitled “Evaluation of Next Generation Sequencing applied to Cryptosporidium parvum and Cryptosporidium hominis epidemiological study” by Dr. Bailly and colleagues, reports on an evaluation of the reliability of Next Generation Sequencing (NGS) as a sequencing tool for cryptosporidiosis, while different applications of the method were also tested. Main results indicate that NGS was useful in investigating Cryptosporidium hominis and parvum genomes, especially for identifying minority variants.

The manuscript might have an adequate impact in the field as it describes important findings on NGS technique upon cryptosporidiosis sequencing in clinical samples. Moreover, the work is also highly informative as a total of 102 clinical samples were tested and sequenced. 

The manuscript is in general well written, clear, concise, and easily to understand; the experimental design is appropriate, while results are well presented and discussed. In my opinion, tche ms can be accepted following a minor revision. Please find few suggestions for making the ms more suitable for publication in Pathogens.

Comments 

SgS should be Sanger sequencing (SgS) when mentioned for the first time

Lines 60-61 the sequencing methods should be, at least briefly, mentioned

Line 80 instead of “conventional PCR” I suggest “qualitative PCR”

Lines 86-87 Various NGS approaches have also been efficiently developed for sequencing and detection of viral genomes, including SARS-CoV-2, as described in detail here https://doi.org/10.3390/microorganisms10061193 and here PMID: 27250973. This information should be included 

Spaces between paragraphs should be removed, e.g., discussion, lines 188, 196, 202, 212, 219, 227, 

If, possible, more supporting references should be included in the methods section as several sentences are lacking in supporting references  

Conclusions should be moved at the end of the discussion

Author Response

Responses to Reviewer 1 comments:

REVISION Manuscript Number: ID-1787620

Dear editor, the authors have considered all the comments of the two reviewers, whom we thank for their constructive annotations. The corrections in the main text are marked up using the “Track Changes” function and a point by point response to the comments has been written in the cover letter (see below). The lines refer to the “clean version” of the manuscript. We hope that our manuscript is now suitable for publication.

Yours sincerely. Pr. F. Dalle.

The manuscript (ID-1787620, pathogens) entitled “Evaluation of Next Generation Sequencing applied to Cryptosporidium parvum and Cryptosporidium hominis epidemiological study” by Dr. Bailly and colleagues, reports on an evaluation of the reliability of Next Generation Sequencing (NGS) as a sequencing tool for cryptosporidiosis, while different applications of the method were also tested. Main results indicate that NGS was useful in investigating Cryptosporidium hominis and parvum genomes, especially for identifying minority variants.

 The manuscript might have an adequate impact in the field as it describes important findings on NGS technique upon cryptosporidiosis sequencing in clinical samples. Moreover, the work is also highly informative as a total of 102 clinical samples were tested and sequenced.

 The manuscript is in general well written, clear, concise, and easily to understand; the experimental design is appropriate, while results are well presented and discussed. In my opinion, the ms can be accepted following a minor revision. Please find few suggestions for making the ms more suitable for publication in Pathogens.

We thank the reviewer for these comments.

SgS should be Sanger sequencing (SgS) when mentioned for the first time.

We thank the reviewer for his remark. As suggested, SgS appear now when “Sanger sequencing” is mentioned for the first time (Line 91).

Lines 60-61 the sequencing methods should be, at least briefly, mentioned.

As the reviewer rightly pointed out, we have mentioned existing techniques, resulting in the following sentence: “These last years, numerous molecular biological tools have been developed to detect and differentiate Cryptosporidium species improving epidemiological knowledge in cryptosporidiosis (1,10,11). These tools include PCR-based genotyping, Sanger sequencing of PCR products, restriction fragment length polymorphism (RFLP) analysis of PCR products, and qPCR assays using fluorescent probes and melting curves analysis (11). Thus, development of molecular techniques have notably allowed to show that, while C. parvum and C. hominis are reported as responsible for 96% of the case of cryptosporidiosis in France and worldwide, at least 20 other Cryptosporidium species are able to infect humans (5,10).“ (Lines 65 to 73). 

Line 80 instead of “conventional PCR” I suggest “qualitative PCR”.

As suggested by the first reviewer, “conventional PCR” has been replaced by “qualitative PCR” (Line 89).

Lines 86-87 Various NGS approaches have also been efficiently developed for sequencing and detection of viral genomes, including SARS-CoV-2, as described in detail here https://doi.org/10.3390/microorganisms10061193 and here PMID: 27250973. This information should be included.

The reviewer highlighted the importance of NGS approaches in other fields of microbiology, notably virology. We agree that this is a point of interest that we have to underline in the manuscript. We thus added the following sentence: “These sequencing approaches have already demonstrated their added value in all fields of microbiology, notably for the sequencing and detection of viral genomes, including SARS-CoV-2.” (Line 97). We also added the corresponding reference in the bibliography:  Rotondo JC, Martini F, Maritati M, et al. Advanced Molecular and Immunological Diagnostic Methods to Detect SARS-CoV-2 Infection. Microorganisms. 2022;10(6):1193. doi:10.3390/microorganisms10061193 (Line 509).

Spaces between paragraphs should be removed, e.g., discussion, lines 188, 196, 202, 212, 219, 227.

We thank the reviewer for this observation. Spaces between those 6 paragraphs of the “Discussion” section have been removed as requested.

If, possible, more supporting references should be included in the methods section as several sentences are lacking in supporting references.

We thank the reviewer for his remark, to which we have tried to answer. Firstly, we have added the references which correspond to the outbreaks of Grasse and Divonne les Bains (Lines 362 and 363). Then, we added a reference related to the choice of primers AL3532 and LX0029 for PCR A and B (Sulaiman IM, Hira PR, Zhou L, et al. Unique endemicity of cryptosporidiosis in children in Kuwait. J Clin Microbiol. 2005;43(6):2805-2809. doi:10.1128/JCM.43.6.2805-2809.2005) (Line 377), which also appears in the bibliography at line 501.

Finally, lines 375 to 394 were devoted to the description of Illumina 16S MSLP protocol, for which we cannot add a reference. However, we made sure to be clearer on the fact that we followed the manufacturer's recommendations - lines 373 and 393

Conclusions should be moved at the end of the discussion.

As requested, conclusions have been moved at the end of the discussion section (Lines 341 to 349).

Reviewer 2 Report

Next generation sequencing has been applied for identification of co-infection of Cryptosporidium species as well as subtypes of each species in various studies. What makes the value of this study was mainly focusing on the identification of minor subtypes by NGS while SgS is unable to achieve it. However, the criteria of an interpretation threshold of NGS data analysis and justification of results for contamination were lacking. Please describe these two points above and provide information for comments within the main text:

Line 52: 20,000 in English. Please revise.

Line 133: 1) The decimal symbol is a period in English. Please revise the notation; 2) The total number of NGS sequences of each sample does not match the sum of sequences from all subtypes. Please explain; 3) Please provide the explanation for the high number (983) of C. parvum subtype in the negative control; 4) Were samples in Table 1 and 2 sequenced in the same run with the negative control since the data of the negative control was identical in Table 1 and 2?

Line 151: The statement depicted the contamination in each sample of the same run. Please explain how you justify the data after the finding of contamination.

Line 155: 1) The decimal symbol is a period in English. Please revise the notation; 2) The total number of NGS sequences of each sample does not match the sum of sequences from all subtypes. Please explain.

Line 164: Please specify the location of the data in Table S3 and explain the discrepancy of the number between the context and Table S3 for sample 2249.

Line 228: Other than the definition of other studies, what is your set up criteira for an interpretation threshold in this study and, how would you differentiate the minorities of C. subtypes from sequencing errors?

Line 258: Please present the NGS data of these 108 DNA samples in the main context instead. This is the strong evidence for comparison of SgS and NGS.

Line 336: Please use English for notation in Tables.

Line 365: 1) Check and revise the information of citations in English; 2) Revise the format of references.

Author Response

Responses to Reviewer 2 comments:

 REVISION Manuscript Number: ID-1787620

Dear editor, the authors have considered all the comments of the two reviewers, whom we thank for their constructive annotations. The corrections in the main text are marked up using the “Track Changes” function and a point by point response to the comments has been written in the cover letter (see below). The lines refer to the “clean version” of the manuscript. We hope that our manuscript is now suitable for publication.

Yours sincerely. Pr. F. Dalle.

 Next generation sequencing has been applied for identification of co-infection of Cryptosporidium species as well as subtypes of each species in various studies. What makes the value of this study was mainly focusing on the identification of minor subtypes by NGS while SgS is unable to achieve it. However, the criteria of an interpretation threshold of NGS data analysis and justification of results for contamination were lacking. Please describe these two points above and provide information for comments within the main text:

We thank the reviewer for this general comment.

Line 52: 20,000 in English. Please revise.

We revised the English notation (line 52). Notation was also revised in all the tables of the manuscript, including those in supplementary data.  We thank the reviewer for allowing us to make these corrections.

Line 133: 1) The decimal symbol is a period in English. Please revise the notation.

The notation has been revised as requested (Line 163 – Table 1).

Line 133: 2) The total number of NGS sequences of each sample does not match the sum of sequences from all subtypes. Please explain.

The reviewer is right, the sum of sequences from all subtypes does not match the total number of NGS sequences showed in the last column of the table. The explanation is that in each NGS run, a small number of sequences were not assigned to any sequence in our database, nor to BLAST. They were therefore "Not assigned". To avoid cluttering the table with an extra column, we did not want to put this information in the table. However, we have omitted to put a sentence below the table mentioning the existence of these ‘’Not assigned’’ sequences. We thank the reviewer for allowing us to correct this error. We propose to add the following sentence: « Some sequences generated during the NGS run could not be assigned to any sequence in our database, nor to BLAST. These "Not Assigned" sequences represented an average of 34.8 (between 0 and 161) sequences in each sample of the run (data not shown) » (Line 156). If the editor prefers that the data regarding the "Not assigned sequences" appear in the table, we can add them.

Line 133: 3) Please provide the explanation for the high number (983) of C. parvum subtype in the negative control.

We propose to add the following sentence. « Two hypotheses could explain the high number of this subtype sequences detected in the negative control: (i) almost half of the samples handled in the run were positive for this subtype, and (ii) as previously described, the prevalence of contaminants can be higher in negative controls than in other samples due to the absence of competitive DNA during the sequencing process [22]. » (Line 148). Furthermore, the reference 22 has been added in the Bibliography section (Line 520): Davis NM, Proctor DM, Holmes SP, Relman DA, Callahan BJ. Simple statistical identification and removal of contaminant sequences in marker-gene and metagenomics data. Microbiome. 2018;6(1):226. doi:10.1186/s40168-018-0605-2.

We understand the reviewer that this high number in the negative control is surprising, but this is what we obtained and presented with complete transparency. Finally, when the two explanations proposed are hypotheses, these observations highlight the sensitivity of these techniques and the high risk of cross contaminations that have a random character.

Line 133: 4) Were samples in Table 1 and 2 sequenced in the same run with the negative control since the data of the negative control was identical in Table 1 and 2?

The reviewer is right; samples of Table 1 and Table 2 were sequenced in the same run, with the same negative control. To be clearer on this point, a sentence has been added at line 363: « To optimize the use of NGS flow cells, 1572, 2055 and mixes A to G were handled in the same run as the Divonne les Bains samples ».

Line 151: The statement depicted the contamination in each sample of the same run. Please explain how you justify the data after the finding of contamination.

We thank the reviewer for this important remark. This study represents a technique evaluation, for which there is currently no standardized interpretation threshold. This is why we chose not to define an interpretation threshold in our “Materials and Methods” section, but rather to propose a threshold based on the observation of the sequencing results from the first samples (i.e. 1572, 2055 and mixes A to G). Thus, by observing a highly probable cross-contamination (C. parvum IIaA18G1R1 and C. parvum IIdA22G1) in all the samples, including the negative control, the use of the negative control for the interpretation of the data appeared indeed unavoidable and mandatory. By choosing to use the highest value found in the negative control (983 sequences), we were able to exclude from all the samples tested the sequences generated by cross-contamination (C. parvum IIaA18G1R1 and C. parvum IIdA22G1). However, as underlined in “Discussion” section (Line 284), interpretation of the data based on the presence of contaminating sequences in the negative controls can lead to loss of sensitivity of the technique, which can make the presence of minor variants difficult to interpret.

To answer the reviewer’s request, we modified the “Results” section from Lines 138 to 156. These modifications include the following sentences.

“As in 1572 and 2055 samples, additional C. parvum subtypes, i.e. C. parvum-IIaA18G1R1 and C. parvum-IIdA22G1, were identified in all of the mixtures, including the negative control. It is of importance to note that all of these sequences corresponded to C. parvum subtypes that were handled in the same run. On the basis of these observations, highlighting a highly probable DNA cross-contamination between samples, the presence of the negative control proved to be essential for the interpretation of the data. Sequencing of negative controls revealed the presence of 1,210 sequences (i.e. average value of the three controls) of which 225 matched with C. parvum-IIcA5G3 and C. hominis-IbA10G2. The most represented subtype in the negative control was C. parvum IIdA22G1 reaching 983 sequences. Two hypotheses could explain the high number of this subtype sequences detected in the negative control: (i) almost half of the samples handled in the run were positive for this subtype, and (ii) as previously described, the prevalence of contaminants can be higher in negative controls than in other samples due to the absence of competitive DNA during the sequencing process. Based on this higher value (983 sequences) observed in the negative control, all sequences generated in lower numbers were thus considered as contaminating sequences. Note however that, while mixes C and D were known to contain 1% and 0.1% of C. hominis IbA10G2 respectively, only 777 and 75 sequences respectively were generated by NGS.”

Line 155: 1) The decimal symbol is a period in English. Please revise the notation

The notation has been revised as requested by the 2nd reviewer.

Line 155: 2) The total number of NGS sequences of each sample does not match the sum of sequences from all subtypes. Please explain.

As seen in Table 1, the sum of sequences from all subtypes does not match the total number of NGS sequences showed in the last column of the Table 2. The explanation is that in each NGS run, a small number of sequences are not assigned to any sequence in our database, nor to BLAST. They are therefore "Not assigned". To avoid cluttering the table with an extra column, we did not want to put this information in the table. However, we have omitted to put a sentence below the table to explain that there were ‘’Not assigned’’ sequences. We thank the reviewer for allowing us to correct this error. We propose to add the following sentence: « Some sequences generated during the NGS run could not be assigned to any sequence in our database, nor to BLAST. These "Not Assigned" sequences represented an average of 45.2 (between 0 and 81) sequences in each sample of the run (data not shown) » (Line 204). If the editor prefers that the data regarding the "Not assigned sequences" appear in the table, we can add them.

Line 164: Please specify the location of the data in Table S3 and explain the discrepancy of the number between the context and Table S3 for sample 2249.

The location of the data has been specified (Line 202), and the discrepancy of the values has been corrected at line 201. We apologize for this typing error that had not been seen during our proofreading.

Line 228: Other than the definition of other studies, what is your set up criteria for an interpretation threshold in this study and, how would you differentiate the minorities of C. subtypes from sequencing errors?

Based on the observation of our sequencing results, we proposed to set an interpretation threshold based on the negative control results. Indeed, considering all the sequences generated during the sequencing reaction, with the risk of identifying subtypes as true whereas they are not, is not acceptable. Furthermore, setting an arbitrary threshold before the run, as described in the scientific literature, does not seem acceptable either. Thus, using the highest number of sequences obtained in the negative control (983 sequences in the case of this run) which depicts the highest level of contamination, we decided that any sequences with a number lower than 983 would be considered as contaminating sequences. However, as underlined in the “Discussion” section (Line 284), interpretation of the data based on the presence of contaminating sequences in the negative controls can lead to loss of sensitivity of the technique, which can make the presence of minor variants difficult to interpret. In the case of this run, the chosen threshold led to the underestimation of minority variants within three samples (D45, mixes C and D).

To answer the reviewer’s request, we modified the “Discussion” section from line 269 to line 288. These modifications include the following paragraph.

“However, limits regarding the interpretation of the data were observed using NGS methods. Indeed, the presence of unexpected sequences in low number in all the samples, including negative controls, highlighted the need for operator expertise and technical care since most of those unexpected sequences were linked to Cryptosporidium species/subtypes present in other samples handled in the same run. The negative controls thus appeared essential for the interpretation of the data, by setting an interpretation threshold based on the highest number of sequences generated in the negative control during each sequencing run. Indeed, this is not acceptable to consider all the sequences generated during a sequencing run, with the risk of identifying subtypes as true whereas they are not. The use of the negative control to set a detection threshold is however questionable since cross-contaminations can lead to misinterpretations. As example for sample D45 and mixes C and D in which, detection of C. hominis IbA10G2 sequences was expected, but for which sequencing generated fewer sequences than in the negative control. These observations indicate that minority variants present at 1% for C. hominis would have been possibly ignored when setting up an interpretation threshold as described in our study.  Finally, these observations highlight the difficulty of setting up a threshold for the interpretation of NGS data and the need for standardization of interpretation thresholds to allow comparison of the results of genotyping studies carried out with the same NGS based tools”.

The reviewer also asked us how we would differentiate sequencing errors from real variants. As mentioned in the “Discussion” section at line 325, the presence of minor subtypes which can not be linked to cross-contamination (as C. parvum IIaA11G2R1) should raise questions about sequencing errors. In the case of this study, only one minor subtype was concerned: C. parvum IIaA11G2R1 found in Grasse outbreak samples. Different arguments suggest that this subtype was not linked to PCR/sequencing error but was a real minor variant.

These arguments are:

(i) MiSeq sequencer has an error rate of 0.9 every 100 bases, which is much lower than other sequencers using different technologies (Stoler N, Nekrutenko A. Sequencing error profiles of Illumina sequencing instruments. NAR Genom Bioinform. 2021;3(1):lqab019.. doi:10.1093/nargab/lqab019), and which seems insufficient to explain the presence of C. parvum IIaA11G2R1. Indeed, this would require a much higher error rate to generate, for example, a C. parvum IIaA11G2R1 instead of a C. parvum IIaA17G1R1, and above all these errors should only concern the microsatellite region, which seems unlikely.

(ii) PCR slippage has already been described in microsatellite regions. However, it seems unlikely that a PCR slippage generated the same subtype in 6 different samples.

(iii) Finally, the bioinformatics pipeline used is supposed to limit the impact of PCR/sequencing errors on the final data. DADA2 pipeline is described as a “correcting Illumina-sequenced amplicon errors” pipeline, which proceeds to the modeling of the sequencing error, supposed to make it possible to distinguish mutant sequences from erroneous sequences. Thus, as described by Callahan et al. in several mock communities, DADA2 identified more real variants and output fewer spurious sequences than other methods.

To be clearer, the following sentences has been added/modified in the “Discussion” section (Lines 323 to 332) “In parallel, cross-contamination could be excluded for some unexpected sequences thanks to original features shared by these subtypes sequences (i.e. IIaA11G2R1). The presence of minor subtypes which can not be linked to cross-contamination (as C. parvum IIaA11G2R1) should raise questions about sequencing errors. Sequencing errors could be linked to (i) PCR polymerase slippage as suggested elsewhere (20), linked to polymerase elongation mistakes in microsatellites regions, (ii) sequencing errors linked to the MiSeq sequencer, which is known to have a 0.9% error rate (ref). However, that random sequencing errors could generate an identical subtype in 6 samples seems highly unlikely. Additionally, the DADA2 bioinformatics pipeline is supposed to limit the impact of PCR/sequencing errors on the final data. This minority subtype could be due to PCR polymerase slippage as suggested elsewhere (20), linked to polymerase elongation mistakes in microsatellites regions. However, since bioinformatics analyses were strict and other Cryptosporidium sp. positive samples of the same run did not contain mixtures, this suggested real mixtures (20). This minority variant could therefore reflect the end of the clonal epidemic wave, or could be linked Another explanation could be within-host genetic diversity, resulting from mutations or recombination during parasitic life cycle (15).“ Furthermore, the following sentences has been added in the “Materials and Methods” section: “This pipeline is supposed to limit the impact of PCR/sequencing errors on the final data. DADA2 is described as a “correcting Illumina-sequenced amplicon errors” pipeline, which proceeds to the modeling of the sequencing error, supposed to make it possible to distinguish mutant sequences from erroneous sequences. Thus, as described by Callahan et al. in several mock communities, DADA2 identified more real variants and output fewer spurious sequences than other methods.“(Lines 405 to 410).

Line 258: Please present the NGS data of these 108 DNA samples in the main context instead. This is the strong evidence for comparison of SgS and NGS.

As rightly underlined by the reviewer, it is important to show the comparison between SgS and NGS more explicitly using a table or a figure. However, the table S4 seemed complicated to us to put in the main context, because of its large size. We thus suggest adding Figure 1 (Line 210), which allows comparing the investigation of Grasse outbreak made by SgS or NGS, in terms of variant proportion and mixed infections detection.

Line 336: Please use English for notation in Tables.

The notation has been revised as requested by the 2nd reviewer.

Line 365: 1) Check and revise the information of citations in English.

We thank the reviewer for this comment. The relevance of the references already present in the original manuscript has been checked, and numbering has been updated as references were added. A total of six references were added in response to the different reviews.

  1. P. Chaud, L. Ramalli, J. Raibaut, C. Ortmans, S. Joubert, F. Chereau, F. Dalle, S. Valot, D. Costa, A. François, Épidémie de cryp-tosporidiose d’origine hydrique dans les Alpes Maritimes–novembre 2019, Médecine et Maladies Infectieuses, Volume 50, Issue 6, Supplement, 2020, Page S167, ISSN 0399-077X doi:10.1016/j.medmal.2020.06.357.

  1. Sulaiman IM, Hira PR, Zhou L, et al. Unique endemicity of cryptosporidiosis in children in Kuwait. J Clin Microbiol. 2005;43(6):2805-2809. doi:10.1128/JCM.43.6.2805-2809.2005

  1. Rotondo JC, Martini F, Maritati M, et al. Advanced Molecular and Immunological Diagnostic Methods to Detect SARS-CoV-2 Infection. Microorganisms. 2022;10(6):1193. doi:10.3390/microorganisms10061193

  1. Davis NM, Proctor DM, Holmes SP, Relman DA, Callahan BJ. Simple statistical identification and removal of contaminant sequences in marker-gene and metagenomics data. Microbiome. 2018;6(1):226. doi:10.1186/s40168-018-0605-2

  1. Ramo A, Quílez J, Del Cacho E, Sánchez-Acedo C. Optimization of a fragment size analysis tool for identification of Cryptosporidium species and Gp60 alleles infecting domestic ruminants. Vet Parasitol. 2014;205(3-4):466-471. doi:10.1016/j.vetpar.2014.08.025

  1. Stoler N, Nekrutenko A. Sequencing error profiles of Illumina sequencing instruments. NAR Genom Bioinform. 2021;3(1):lqab019.. doi:10.1093/nargab/lqab019

Line 365: 2) Revise the format of references.

As suggested, the reference format has been homogenized and adapted to the recommendations of the "instructions for authors" section of Pathogens Journal website.

Round 2

Reviewer 2 Report

Thank you for the revision.

Please note that contamination is a great issue for studies. Please see comments and provide more information for clarifying contamination issue in the negative control. After confirmation and clarification, this study would be more valuable.

Author Response

REVISION Manuscript Number: ID-1787620

Dear editor, the authors have considered all the comments of the reviewer 2. The corrections in the main text are marked up using the “Track Changes” function and a point by point response to the comments has been written in the cover letter (see below). The lines refer to the “clean version” of the manuscript. We hope that our manuscript is now suitable for publication.

Yours sincerely. Pr. F. Dalle.

Next generation sequencing has been applied for identification of co-infection of Cryptosporidium species as well as subtypes of each species in various studies. What makes the value of this study was mainly focusing on the identification of minor subtypes by NGS while SgS is unable to achieve it. However, the criteria of an interpretation threshold of NGS data analysis and justification of results for contamination were lacking. Please describe these two points above and provide information for comments within the main text:

We thank the reviewer for this general comment.

Line 52: 20,000 in English. Please revise.

We revised the English notation (line 52). Notation was also revised in all the tables of the manuscript, including those in supplementary data.  We thank the reviewer for allowing us to make these corrections.

Response: Noted and appreciated.

Answer: Ok

Line 133: 1) The decimal symbol is a period in English. Please revise the notation.

The notation has been revised as requested (Line 163 – Table 1).

Response: Noted and appreciated.

Answer: Ok

Line 133: 2) The total number of NGS sequences of each sample does not match the sum of sequences from all subtypes. Please explain.

The reviewer is right, the sum of sequences from all subtypes does not match the total number of NGS sequences showed in the last column of the table. The explanation is that in each NGS run, a small number of sequences were not assigned to any sequence in our database, nor to BLAST. They were therefore "Not assigned". To avoid cluttering the table with an extra column, we did not want to put this information in the table. However, we have omitted to put a sentence below the table mentioning the existence of these ‘’Not assigned’’ sequences. We thank the reviewer for allowing us to correct this error. We propose to add the following sentence: « Some sequences generated during the NGS run could not be assigned to any sequence in our database, nor to BLAST. These "Not Assigned" sequences represented an average of 34.8 (between 0 and 161) sequences in each sample of the run (data not shown) » (Line 156). If the editor prefers that the data regarding the "Not assigned sequences" appear in the table, we can add them.

Response: The addition of the description is noted and appreciated. In order to keep the integrity of the data, please add the number of the “not assigned” sequences to Table 1 (Suggestion: number of “not assigned” sequences can be in the parenthesis following the total number in the same column).

Answer: We thank the reviewer for his suggestion and that has been done. The mention of the “not assigned” sequences appears now, as suggested, in the same column as the “total number of sequences” (Line 163). The following sentence appears line 157: “Some sequences generated during the NGS run could not be assigned to any sequence in our database, nor to BLAST (Table 1)”.

 Line 133: 3) Please provide the explanation for the high number (983) of C. parvum subtype in the negative control.

We propose to add the following sentence. « Two hypotheses could explain the high number of this subtype sequences detected in the negative control: (i) almost half of the samples handled in the run were positive for this subtype, and (ii) as previously described, the prevalence of contaminants can be higher in negative controls than in other samples due to the absence of competitive DNA during the sequencing process [22]. » (Line 148). Furthermore, the reference 22 has been added in the Bibliography section (Line 520): Davis NM, Proctor DM, Holmes SP, Relman DA, Callahan BJ. Simple statistical identification and removal of contaminant sequences in marker-gene and metagenomics data. Microbiome2018;6(1):226. doi:10.1186/s40168-018-0605-2.

We understand the reviewer that this high number in the negative control is surprising, but this is what we obtained and presented with complete transparency. Finally, when the two explanations proposed are hypotheses, these observations highlight the sensitivity of these techniques and the high risk of cross contaminations that have a random character.

Response:

Addition of reference 22 is noted and appreciated. The whole data set would be unreliable if there is no clear description after the finding of contamination. Contamination should be fairly described for data explanation. Please provide the following information: 1) provide evidence with experiments to confirm and clarify whether there was contamination in the negative control of this study.

Answer: 1) As explained in the manuscript line 400, we used negative controls in all our experiments to check for the presence of contaminations. Thus, all sequences generated by NGS in our negative controls were interpreted as contamination. This methodology is commonly employed in molecular biology experiments. DeMone et al., when studying NGS applied to Cryptosporidium sp. interpreted the presence of sequences in the negative control, like us, as contaminations (DeMone C, Hwang MH, Feng Z, et al. Application of next generation sequencing for detection of protozoan pathogens in shell-fish. Food Waterborne Parasitol. 2020;21:e00096. doi:10.1016/j.fawpar.2020.e00096). Moreover, before applying NGS to Cryptosporidium sp., during the testing/development step of the technique in our laboratory, we carried out a run composed only of blanks, which showed the absence of Cryptosporidium sp. sequences. Such an experiment comforted us with the absence of technique-related artifacts (e.g. absence of contaminations in the reagents), and that the presence of Cryptosporidium sp. sequences in the current study was related to contaminations originating from Cryptosporidium species/genotypes handled in the on-going NGS run (see below, answer to point 2).

2) If contamination exists, please identify what kind of contamination in the negative control (cross contamination, reagents, etc.) of this study.

Answer: 2) As explained above, we are firmly convinced that the negative controls attest to the presence of contaminations in our experiments. Several arguments strongly suggest that these contaminations originate from DNA of the samples handled within the same NGS run:

- Firstly, as mentioned in the manuscript lines 141 and 202, all the sequences found in the negative control corresponded to Cryptosporidium subtypes handled in the run. Moreover, the proportion of Cryptosporidium subtype sequences found in the negative control deals with the proportion of positive samples for those subtypes within the run, as illustrated in the following table:  

- Secondly, since this study, we performed others NGS experiments and we observed the same phenomenon, namely the presence in the negative control of Cryptosporidium subtypes handled within the run (data unpublished). Thus, the Cryptosporidium subtypes observed in the negative control differ from one experiment to another, according to the samples manipulated in the run, arguing a cross-contamination process.

-  In addition, the reagents having been changed between each run and the subtypes having been found in the negative control differing between each experiment (unpublished data), this excludes contamination of the reagents. Indeed, there is a very low probability that several batches of different reagents would be contaminated each time by the Cryptosporidium subtypes manipulated in the corresponding run.

- Finally, the contamination does not seem to come from the laboratory environment. Indeed, such contaminations were observed during our first tests with the presence of sequences attributed to HIV, bacteria, SARS-CoV-2 and human DNA in all the samples of the runs. We thus decided to improve our practices with (i) the use of a dedicated room, separated from virology and bacteriology laboratories, (ii) the use of dedicated equipment (pipettes, pipette tips, …), and (iii) the use of a dedicated decontamination process using in particular a work chamber equipped with a UV ramp. The following runs did not generate such sequences anymore, confirming the efficacy of the measures.

Line 152: Please specify the term "lower numbers" of this statement regarding the definition of contamination in this experiment.

Answer: We thank the reviewer for this comment. To be clearer, we modified the following paragraph (Line 148):

The most represented subtype in the negative control was C. parvum IIdA22G1 reaching 983 sequences. Two hypotheses could explain the high number of this subtype sequences detected in the negative controls: (i) almost half of the samples handled in the run were positive for this subtype, and (ii) as previously described, the prevalence of contaminants can be higher in negative controls than in other samples due to the absence of competitive DNA during the sequencing process [22]. Based on this higher value (983 sequences) observed in the negative control, all sequences generated in lower numbers were thus considered contamination. This highest value observed in the negative control was thus used as the interpretation threshold. Thereby, any sequence generated by NGS with 983 sequences or less was considered a contaminant.

Line 155: Regarding sequences of C. hominis IbA10G2, it was 75 sequences in sample mix D while it was 131 sequences in negative control. Please provide the explanation if this result of mix D was reliable. One can argue that it was 1140 sequences of C. parvum_IIcA5G3 in sample mix G while it also accounted for 0.1% of DNA in the sample. One possible way is to prove that the mix D is indeed the mixture of DNA with 99.9% from C. parvum_IIcA5G3 and 0.1% from C. hominis_IbA10G2 via alternative approaches with negative control in the experiment.

Answer: As already described in other publications, notably from Costa et al. (Costa D, Soulieux L, Razakandrainibe R, et al. Comparative Performance of Eight PCR Methods to Detect Cryptosporidium Species. Pathogens. 2021;10(6):647.  doi:10.3390/pathogens10060647) and Valeix et al. (Valeix N, Costa D, Basmaciyan L, et al. Multicenter Comparative Study of Six Cryptosporidium parvum DNA Extraction Proto-cols Including Mechanical Pretreatment from Stool Samples. Microorganisms. 2020;8(9):1450. doi:10.3390/microorganisms8091450), there is a preferential amplification of C. parvum DNA compared to C. hominis DNA by PCR methods. This therefore implies a better detection threshold for C. parvum than for C. hominis in these different studies. It seems that we observed the same phenomenon in our study with NGS. Indeed, by observing the mixtures A to G, in equal proportion, C. parvum generated more sequences by NGS than C. hominis.To be clearer on this point in the manuscript, we added the following sentence in the Discussion (line 255): “Previous studies have shown that a preferential amplification of C. parvum DNA compared to C. hominis DNA exists by PCR methods [30]. This certainly explains that, in equal proportion, C. parvum generated more sequences by NGS than C. hominis in the mixtures A to G. “

Line 133: 4) Were samples in Table 1 and 2 sequenced in the same run with the negative control since the data of the negative control was identical in Table 1 and 2?

The reviewer is right; samples of Table 1 and Table 2 were sequenced in the same run, with the same negative control. To be clearer on this point, a sentence has been added at line 363: « To optimize the use of NGS flow cells, 1572, 2055 and mixes A to G were handled in the same run as the Divonne les Bains samples ».

Response: Noted and appreciated.

Answer: Ok

Line 151: The statement depicted the contamination in each sample of the same run. Please explain how you justify the data after the finding of contamination.

We thank the reviewer for this important remark. This study represents a technique evaluation, for which there is currently no standardized interpretation threshold. This is why we chose not to define an interpretation threshold in our “Materials and Methods” section, but rather to propose a threshold based on the observation of the sequencing results from the first samples (i.e. 1572, 2055 and mixes A to G). Thus, by observing a highly probable cross-contamination (C. parvum IIaA18G1R1 and C. parvum IIdA22G1) in all the samples, including the negative control, the use of the negative control for the interpretation of the data appeared indeed unavoidable and mandatory. By choosing to use the highest value found in the negative control (983 sequences), we were able to exclude from all the samples tested the sequences generated by cross-contamination (C. parvum IIaA18G1R1 and C. parvum IIdA22G1). However, as underlined in “Discussion” section (Line 284), interpretation of the data based on the presence of contaminating sequences in the negative controls can lead to loss of sensitivity of the technique, which can make the presence of minor variants difficult to interpret.

Response: Explanation appreciated. However, it seems contamination did exist in the negative control but still not be confirmed yet. Please see comments above to provide information for confirmation of contamination in the negative control of this study.

Answer: As requested by the reviewer, we answered to the comments above.

After confirmation, the discovery and discussion of this study would be more valuable.

To answer the reviewer’s request, we modified the “Results” section from Lines 138 to 156. These modifications include the following sentences.

“As in 1572 and 2055 samples, additional C. parvum subtypes, i.e. C. parvum-IIaA18G1R1 and C. parvum-IIdA22G1, were identified in all of the mixtures, including the negative control. It is of importance to note that all of these sequences corresponded to C. parvum subtypes that were handled in the same run. On the basis of these observations, highlighting a highly probable DNA cross-contamination between samples, the presence of the negative control proved to be essential for the interpretation of the data. Sequencing of negative controls revealed the presence of 1,210 sequences (i.e. average value of the three controls) of which 225 matched with C. parvum-IIcA5G3 and C. hominis-IbA10G2. The most represented subtype in the negative control was C. parvum IIdA22G1 reaching 983 sequences. Two hypotheses could explain the high number of this subtype sequences detected in the negative control: (i) almost half of the samples handled in the run were positive for this subtype, and (ii) as previously described, the prevalence of contaminants can be higher in negative controls than in other samples due to the absence of competitive DNA during the sequencing process. Based on this higher value (983 sequences) observed in the negative control, all sequences generated in lower numbers were thus considered as contaminating sequences. Note however that, while mixes C and D were known to contain 1% and 0.1% of C. hominis IbA10G2 respectively, only 777 and 75 sequences respectively were generated by NGS.”

Response: Noted and appreciated. Please see response above for Line 152 and Line 155.

Answer: As requested by the reviewer, we answered to the comments above.

Line 155: 1) The decimal symbol is a period in English. Please revise the notation

The notation has been revised as requested by the 2nd reviewer.

Response: Noted and appreciated. 

Answer: Ok

Line 155: 2) The total number of NGS sequences of each sample does not match the sum of sequences from all subtypes. Please explain.

As seen in Table 1, the sum of sequences from all subtypes does not match the total number of NGS sequences showed in the last column of the Table 2. The explanation is that in each NGS run, a small number of sequences are not assigned to any sequence in our database, nor to BLAST. They are therefore "Not assigned". To avoid cluttering the table with an extra column, we did not want to put this information in the table. However, we have omitted to put a sentence below the table to explain that there were ‘’Not assigned’’ sequences. We thank the reviewer for allowing us to correct this error. We propose to add the following sentence: « Some sequences generated during the NGS run could not be assigned to any sequence in our database, nor to BLAST. These "Not Assigned" sequences represented an average of 45.2 (between 0 and 81) sequences in each sample of the run (data not shown) » (Line 204). If the editor prefers that the data regarding the "Not assigned sequences" appear in the table, we can add them.

Response: The addition of the description is noted and appreciated. In order to keep the integrity of the data, please add the number of the “not assigned” sequences to Table 2 (Suggestion: number of “not assigned” sequences can be in the parenthesis following the total number in the same column).

Answer: We thank the reviewer for this suggestion and that has been done. The mention of the “not assigned” sequences appears now, as suggested, in the same column as the “total number of sequences” (Line 190). The following sentence appears line 204: “Some sequences generated during the NGS run could not be assigned to any sequence in our database, nor to BLAST (Table 2)”.

Line 164: Please specify the location of the data in Table S3 and explain the discrepancy of the number between the context and Table S3 for sample 2249.

The location of the data has been specified (Line 202), and the discrepancy of the values has been corrected at line 201. We apologize for this typing error that had not been seen during our proofreading.

Response: Noted and appreciated. 

Answer: Ok

Line 228: Other than the definition of other studies, what is your set up criteria for an interpretation threshold in this study and, how would you differentiate the minorities of C. subtypes from sequencing errors?

Based on the observation of our sequencing results, we proposed to set an interpretation threshold based on the negative control results. Indeed, considering all the sequences generated during the sequencing reaction, with the risk of identifying subtypes as true whereas they are not, is not acceptable. Furthermore, setting an arbitrary threshold before the run, as described in the scientific literature, does not seem acceptable either. Thus, using the highest number of sequences obtained in the negative control (983 sequences in the case of this run) which depicts the highest level of contamination, we decided that any sequences with a number lower than 983 would be considered as contaminating sequences. However, as underlined in the “Discussion” section (Line 284), interpretation of the data based on the presence of contaminating sequences in the negative controls can lead to loss of sensitivity of the technique, which can make the presence of minor variants difficult to interpret. In the case of this run, the chosen threshold led to the underestimation of minority variants within three samples (D45, mixes C and D).

To answer the reviewer’s request, we modified the “Discussion” section from line 269 to line 288. These modifications include the following paragraph.

“However, limits regarding the interpretation of the data were observed using NGS methods. Indeed, the presence of unexpected sequences in low number in all the samples, including negative controls, highlighted the need for operator expertise and technical care since most of those unexpected sequences were linked to Cryptosporidium species/subtypes present in other samples handled in the same run. The negative controls thus appeared essential for the interpretation of the data, by setting an interpretation threshold based on the highest number of sequences generated in the negative control during each sequencing run. Indeed, this is not acceptable to consider all the sequences generated during a sequencing run, with the risk of identifying subtypes as true whereas they are not. The use of the negative control to set a detection threshold is however questionable since cross-contaminations can lead to misinterpretations. As example for sample D45 and mixes C and D in which, detection of C. hominis IbA10G2 sequences was expected, but for which sequencing generated fewer sequences than in the negative control. These observations indicate that minority variants present at 1% for C. hominis would have been possibly ignored when setting up an interpretation threshold as described in our study.  Finally, these observations highlight the difficulty of setting up a threshold for the interpretation of NGS data and the need for standardization of interpretation thresholds to allow comparison of the results of genotyping studies carried out with the same NGS based tools”.

The reviewer also asked us how we would differentiate sequencing errors from real variants. As mentioned in the “Discussion” section at line 325, the presence of minor subtypes which can not be linked to cross-contamination (as C. parvum IIaA11G2R1) should raise questions about sequencing errors. In the case of this study, only one minor subtype was concerned: C. parvum IIaA11G2R1 found in Grasse outbreak samples. Different arguments suggest that this subtype was not linked to PCR/sequencing error but was a real minor variant.

These arguments are:

(i) MiSeq sequencer has an error rate of 0.9 every 100 bases, which is much lower than other sequencers using different technologies (Stoler N, Nekrutenko A. Sequencing error profiles of Illumina sequencing instruments. NAR Genom Bioinform2021;3(1):lqab019.. doi:10.1093/nargab/lqab019), and which seems insufficient to explain the presence of C. parvum IIaA11G2R1. Indeed, this would require a much higher error rate to generate, for example, a C. parvum IIaA11G2R1 instead of a C. parvum IIaA17G1R1, and above all these errors should only concern the microsatellite region, which seems unlikely.

(ii) PCR slippage has already been described in microsatellite regions. However, it seems unlikely that a PCR slippage generated the same subtype in 6 different samples.

(iii) Finally, the bioinformatics pipeline used is supposed to limit the impact of PCR/sequencing errors on the final data. DADA2 pipeline is described as a “correcting Illumina-sequenced amplicon errors” pipeline, which proceeds to the modeling of the sequencing error, supposed to make it possible to distinguish mutant sequences from erroneous sequences. Thus, as described by Callahan et al. in several mock communities, DADA2 identified more real variants and output fewer spurious sequences than other methods.

To be clearer, the following sentences have been added/modified in the “Discussion” section (Lines 323 to 332) “In parallel, cross-contamination could be excluded for some unexpected sequences thanks to original features shared by these subtypes sequences (i.e. IIaA11G2R1). The presence of minor subtypes which can not be linked to cross-contamination (as C. parvum IIaA11G2R1) should raise questions about sequencing errors. Sequencing errors could be linked to (i) PCR polymerase slippage as suggested elsewhere (20), linked to polymerase elongation mistakes in microsatellites regions, (ii) sequencing errors linked to the MiSeq sequencer, which is known to have a 0.9% error rate (ref). However, that random sequencing errors could generate an identical subtype in 6 samples seems highly unlikely. Additionally, the DADA2 bioinformatics pipeline is supposed to limit the impact of PCR/sequencing errors on the final data. This minority subtype could be due to PCR polymerase slippage as suggested elsewhere (20), linked to polymerase elongation mistakes in microsatellites regions. However, since bioinformatics analyses were strict and other Cryptosporidium sp. positive samples of the same run did not contain mixtures, this suggested real mixtures (20). This minority variant could therefore reflect the end of the clonal epidemic wave, or could be linked Another explanation could be within-host genetic diversity, resulting from mutations or recombination during parasitic life cycle (15).“ Furthermore, the following sentences have been added in the “Materials and Methods” section: “This pipeline is supposed to limit the impact of PCR/sequencing errors on the final data. DADA2 is described as a “correcting Illumina-sequenced amplicon errors” pipeline, which proceeds to the modeling of the sequencing error, supposed to make it possible to distinguish mutant sequences from erroneous sequences. Thus, as described by Callahan et al. in several mock communities, DADA2 identified more real variants and output fewer spurious sequences than other methods.“(Lines 405 to 410).

Response: Noted and appreciated.

Answer: Ok

The suspension of contamination in the negative control seems to be the major hurdle of setting up the threshold in this study. Please see comments above to provide information for confirmation of contamination.

Answer: As requested by the reviewer, we answered to the comments above.

Line 258: Please present the NGS data of these 108 DNA samples in the main context instead. This is the strong evidence for comparison of SgS and NGS.

As rightly underlined by the reviewer, it is important to show the comparison between SgS and NGS more explicitly using a table or a figure. However, the table S4 seemed complicated to us to put in the main context, because of its large size. We thus suggest adding Figure 1 (Line 210), which allows comparing the investigation of Grasse outbreak made by SgS or NGS, in terms of variant proportion and mixed infections detection.

Response: Addition of Figure 1 is noted and appreciated. Please also add the statement from the comparison provided in Figure 1. 

Answer: As requested, the statement from the comparison has been added line 208, with the following sentences: “Figure 1 compares the investigation of the outbreak carried out by SgS and by NGS, in terms of subtypes proportion and mixed infections detection.  By investigating more samples, NGS made it possible to reflect the true proportions of each variant within the outbreak, notably by showing that C. parvum IIdA22G1 represented almost 90% of the subtypes found during the outbreak. Furthermore, NGS was able to highlight mixed infections within the outbreak which had been ignored by SgS”.

Line 336: Please use English for notation in Tables.

The notation has been revised as requested by the 2nd reviewer.

Response: Noted and appreciated. 

Answer: Ok

Line 365: 1) Check and revise the information of citations in English.

We thank the reviewer for this comment. The relevance of the references already present in the original manuscript has been checked, and numbering has been updated as references were added. A total of six references were added in response to the different reviews.

  1. P. Chaud, L. Ramalli, J. Raibaut, C. Ortmans, S. Joubert, F. Chereau, F. Dalle, S. Valot, D. Costa, A. François, Épidémie de cryp-tosporidiose d’origine hydrique dans les Alpes Maritimes–novembre 2019, Médecine et Maladies Infectieuses, Volume 50, Issue 6, Supplement, 2020, Page S167, ISSN 0399-077X doi:10.1016/j.medmal.2020.06.357.

  1. Sulaiman IM, Hira PR, Zhou L, et al. Unique endemicity of cryptosporidiosis in children in Kuwait. J Clin Microbiol. 2005;43(6):2805-2809. doi:10.1128/JCM.43.6.2805-2809.2005

  1. Rotondo JC, Martini F, Maritati M, et al. Advanced Molecular and Immunological Diagnostic Methods to Detect SARS-CoV-2 Infection. Microorganisms. 2022;10(6):1193. doi:10.3390/microorganisms10061193

  1. Davis NM, Proctor DM, Holmes SP, Relman DA, Callahan BJ. Simple statistical identification and removal of contaminant sequences in marker-gene and metagenomics data. Microbiome. 2018;6(1):226. doi:10.1186/s40168-018-0605-2

  1. Ramo A, Quílez J, Del Cacho E, Sánchez-Acedo C. Optimization of a fragment size analysis tool for identification of Cryptosporidium species and Gp60 alleles infecting domestic ruminants. Vet Parasitol. 2014;205(3-4):466-471. doi:10.1016/j.vetpar.2014.08.025

  1. Stoler N, Nekrutenko A. Sequencing error profiles of Illumina sequencing instruments. NAR Genom Bioinform. 2021;3(1):lqab019.. doi:10.1093/nargab/lqab019

 Response: Noted and appreciated.

Answer: Ok

Line 365: 2) Revise the format of references.

As suggested, the reference format has been homogenized and adapted to the recommendations of the "instructions for authors" section of Pathogens Journal website.

Response: Noted and appreciated.

Answer: Ok

Round 3

Reviewer 2 Report

Dear authors,

Comments for explanation and references were appreciated and accepted.

Please find 4 minor typos, highlighted with note, for revision in the manuscript.

Author Response

REVISION Manuscript Number: ID-1787620

Dear editor, the authors have considered all the comments of the reviewer 2. The corrections in the main text are marked up using the “Track Changes” function and a point by point response to the comments has been written in the cover letter (see below). The lines refer to the “clean version” of the manuscript. We hope that our manuscript is now suitable for publication.

Yours sincerely. Pr. F. Dalle.

Next generation sequencing has been applied for identification of co-infection of Cryptosporidium species as well as subtypes of each species in various studies. What makes the value of this study was mainly focusing on the identification of minor subtypes by NGS while SgS is unable to achieve it. However, the criteria of an interpretation threshold of NGS data analysis and justification of results for contamination were lacking. Please describe these two points above and provide information for comments within the main text:

We thank the reviewer for this general comment.

Line 52: 20,000 in English. Please revise.

We revised the English notation (line 52). Notation was also revised in all the tables of the manuscript, including those in supplementary data. We thank the reviewer for allowing us to make these corrections.

Response: Noted and appreciated.

Answer: Ok

Line 133: 1) The decimal symbol is a period in English. Please revise the notation.

The notation has been revised as requested (Line 163 – Table 1).

Response: Noted and appreciated.

Answer: Ok

Line 133: 2) The total number of NGS sequences of each sample does not match the sum of sequences from all subtypes. Please explain.

The reviewer is right, the sum of sequences from all subtypes does not match the total number of NGS sequences showed in the last column of the table. The explanation is that in each NGS run, a small number of sequences were not assigned to any sequence in our database, nor to BLAST. They were therefore "Not assigned". To avoid cluttering the table with an extra column, we did not want to put this information in the table. However, we have omitted to put a sentence below the table mentioning the existence of these ‘’Not assigned’’ sequences. We thank the reviewer for allowing us to correct this error. We propose to add the following sentence: « Some sequences generated during the NGS run could not be assigned to any sequence in our database, nor to BLAST. These "Not Assigned" sequences represented an average of 34.8 (between 0 and 161) sequences in each sample of the run (data not shown) » (Line 156). If the editor prefers that the data regarding the "Not assigned sequences" appear in the table, we can add them.

Response: The addition of the description is noted and appreciated. In order to keep the integrity of the data, please add the number of the “not assigned” sequences to Table 1 (Suggestion: number of “not assigned” sequences can be in the parenthesis following the total number in the same column).

Answer: We thank the reviewer for his suggestion and that has been done. The mention of the “not assigned” sequences appears now, as suggested, in the same column as the “total number of sequences” (Line 163). The following sentence appears line 157: “Some sequences generated during the NGS run could not be assigned to any sequence in our database, nor to BLAST (Table 1)”.

Response: Noted and appreciated.

Answer: We thank the reviewer

Line 133: 3) Please provide the explanation for the high number (983) of C. parvum subtype in the negative control.

We propose to add the following sentence. « Two hypotheses could explain the high number of this subtype sequences detected in the negative control: (i) almost half of the samples handled in the run were positive for this subtype, and (ii) as previously described, the prevalence of contaminants can be higher in negative controls than in other samples due to the absence of competitive DNA during the sequencing process [22]. » (Line 148). Furthermore, the reference 22 has been added in the Bibliography section (Line 520): Davis NM, Proctor DM, Holmes SP, Relman DA, Callahan BJ. Simple statistical identification and removal of contaminant sequences in marker-gene and metagenomics data. Microbiome2018;6(1):226. doi:10.1186/s40168-018-0605-2.

We understand the reviewer that this high number in the negative control is surprising, but this is what we obtained and presented with complete transparency. Finally, when the two explanations proposed are hypotheses, these observations highlight the sensitivity of these techniques and the high risk of cross contaminations that have a random character.

Response:

Addition of reference 22 is noted and appreciated. The whole data set would be unreliable if there is no clear description after the finding of contamination. Contamination should be fairly described for data explanation. Please provide the following information: 1) provide evidence with experiments to confirm and clarify whether there was contamination in the negative control of this study.

Answer: 1) As explained in the manuscript line 400, we used negative controls in all our experiments to check for the presence of contaminations. Thus, all sequences generated by NGS in our negative controls were interpreted as contamination. This methodology is commonly employed in molecular biology experiments. DeMone et al., when studying NGS applied to Cryptosporidium sp. interpreted the presence of sequences in the negative control, like us, as contaminations (DeMone C, Hwang MH, Feng Z, et al. Application of next generation sequencing for detection of protozoan pathogens in shell-fish. Food Waterborne Parasitol. 2020;21:e00096. doi:10.1016/j.fawpar.2020.e00096). Moreover, before applying NGS to Cryptosporidium sp., during the testing/development step of the technique in our laboratory, we carried out a run composed only of blanks, which showed the absence of Cryptosporidium sp. sequences. Such an experiment comforted us with the absence of technique-related artifacts (e.g. absence of contaminations in the reagents), and that the presence of Cryptosporidium sp. sequences in the current study was related to contaminations originating from Cryptosporidium species/genotypes handled in the on-going NGS run (see below, answer to point 2).

Response: 1) Reference and clarification were appreciated.

Answer: We thank the reviewer

2) If contamination exists, please identify what kind of contamination in the negative control (cross contamination, reagents, etc.) of this study.

Answer: 2) As explained above, we are firmly convinced that the negative controls attest to the presence of contaminations in our experiments. Several arguments strongly suggest that these contaminations originate from DNA of the samples handled within the same NGS run:

- Firstly, as mentioned in the manuscript lines 141 and 202, all the sequences found in the negative control corresponded to Cryptosporidium subtypes handled in the run. Moreover, the proportion of Cryptosporidium subtype sequences found in the negative control deals with the proportion of positive samples for those subtypes within the run, as illustrated in the following table:

- Secondly, since this study, we performed others NGS experiments and we observed the same phenomenon, namely the presence in the negative control of Cryptosporidium subtypes handled within the run (data unpublished). Thus, the Cryptosporidium subtypes observed in the negative control differ from one experiment to another, according to the samples manipulated in the run, arguing a cross-contamination process.

- In addition, the reagents having been changed between each run and the subtypes having been found in the negative control differing between each experiment (unpublished data), this excludes contamination of the reagents. Indeed, there is a very low probability that several batches of different reagents would be contaminated each time by the Cryptosporidium subtypes manipulated in the corresponding run.

- Finally, the contamination does not seem to come from the laboratory environment. Indeed, such contaminations were observed during our first tests with the presence of sequences attributed to HIV, bacteria, SARS-CoV-2 and human DNA in all the samples of the runs. We thus decided to improve our practices with (i) the use of a dedicated room, separated from virology and bacteriology laboratories, (ii) the use of dedicated equipment (pipettes, pipette tips, …), and (iii) the use of a dedicated decontamination process using in particular a work chamber equipped with a UV ramp. The following runs did not generate such sequences anymore, confirming the efficacy of the measures.

Response: 2) Explanation was appreciated.

Answer: We thank the reviewer

Line 152: Please specify the term "lower numbers" of this statement regarding the definition of contamination in this experiment.

Answer: We thank the reviewer for this comment. To be clearer, we modified the following paragraph (Line 148):

The most represented subtype in the negative control was C. parvum IIdA22G1 reaching 983 sequences. Two hypotheses could explain the high number of this subtype sequences detected in the negative controls: (i) almost half of the samples handled in the run were positive for this subtype, and (ii) as previously described, the prevalence of contaminants can be higher in negative controls than in other samples due to the absence of competitive DNA during the sequencing process [22]. Based on this higher value (983 sequences) observed in the negative control, all sequences generated in lower numbers were thus considered contamination. This highest value observed in the negative control was thus used as the interpretation threshold. Thereby, any sequence generated by NGS with 983 sequences or less was considered a contaminant.

Response: Revision was noted and appreciated.

Answer: We thank the reviewer

Line 155: Regarding sequences of C. hominis IbA10G2, it was 75 sequences in sample mix D while it was 131 sequences in negative control. Please provide the explanation if this result of mix D was reliable. One can argue that it was 1140 sequences of C. parvum_IIcA5G3 in sample mix G while it also accounted for 0.1% of DNA in the sample. One possible way is to prove that the mix D is indeed the mixture of DNA with 99.9% from C. parvum_IIcA5G3 and 0.1% from C. hominis_IbA10G2 via alternative approaches with negative control in the experiment.

Answer: As already described in other publications, notably from Costa et al. (Costa D, Soulieux L, Razakandrainibe R, et al. Comparative Performance of Eight PCR Methods to Detect Cryptosporidium Species. Pathogens. 2021;10(6):647. doi:10.3390/pathogens10060647) and Valeix et al. (Valeix N, Costa D, Basmaciyan L, et al. Multicenter Comparative Study of Six Cryptosporidium parvum DNA Extraction Proto-cols Including Mechanical Pretreatment from Stool Samples. Microorganisms2020;8(9):1450. doi:10.3390/microorganisms8091450)there is a preferential amplification of C. parvum DNA compared to C. hominis DNA by PCR methods. This therefore implies a better detection threshold for C. parvum than for C. hominis in these different studies. It seems that we observed the same phenomenon in our study with NGS. Indeed, by observing the mixtures A to G, in equal proportion, C. parvum generated more sequences by NGS than C. hominis.To be clearer on this point in the manuscript, we added the following sentence in the Discussion (line 255): “Previous studies have shown that a preferential amplification of C. parvum DNA compared to C. hominis DNA exists by PCR methods [30]. This certainly explains that, in equal proportion, C. parvum generated more sequences by NGS than C. hominis in the mixtures A to G. “

Response: Noted and appreciated. Please consider to cite the article of Pathogens. 2021;10(6):647, instead of Ref 30 for this statement if appropriate. 

Answer: We thank the reviewer. The reference 30 was replaced as suggested by the reviewer

Line 133: 4) Were samples in Table 1 and 2 sequenced in the same run with the negative control since the data of the negative control was identical in Table 1 and 2?

The reviewer is right; samples of Table 1 and Table 2 were sequenced in the same run, with the same negative control. To be clearer on this point, a sentence has been added at line 363: « To optimize the use of NGS flow cells, 1572, 2055 and mixes A to G were handled in the same run as the Divonne les Bains samples ».

Response: Noted and appreciated.

Answer: Ok

Line 151: The statement depicted the contamination in each sample of the same run. Please explain how you justify the data after the finding of contamination.

We thank the reviewer for this important remark. This study represents a technique evaluation, for which there is currently no standardized interpretation threshold. This is why we chose not to define an interpretation threshold in our “Materials and Methods” section, but rather to propose a threshold based on the observation of the sequencing results from the first samples (i.e. 1572, 2055 and mixes A to G). Thus, by observing a highly probable cross-contamination (C. parvum IIaA18G1R1 and C. parvum IIdA22G1) in all the samples, including the negative control, the use of the negative control for the interpretation of the data appeared indeed unavoidable and mandatory. By choosing to use the highest value found in the negative control (983 sequences), we were able to exclude from all the samples tested the sequences generated by cross-contamination (C. parvum IIaA18G1R1 and C. parvum IIdA22G1). However, as underlined in “Discussion” section (Line 284), interpretation of the data based on the presence of contaminating sequences in the negative controls can lead to loss of sensitivity of the technique, which can make the presence of minor variants difficult to interpret.

Response: Explanation appreciated. However, it seems contamination did exist in the negative control but still not be confirmed yet. Please see comments above to provide information for confirmation of contamination in the negative control of this study.

Answer: As requested by the reviewer, we answered to the comments above.

Response: Explanation was noted and appreciated.

Answer: We thank the reviewer

After confirmation, the discovery and discussion of this study would be more valuable.

To answer the reviewer’s request, we modified the “Results” section from Lines 138 to 156. These modifications include the following sentences.

“As in 1572 and 2055 samples, additional C. parvum subtypes, i.e. C. parvum-IIaA18G1R1 and C. parvum-IIdA22G1, were identified in all of the mixtures, including the negative control. It is of importance to note that all of these sequences corresponded to C. parvum subtypes that were handled in the same run. On the basis of these observations, highlighting a highly probable DNA cross-contamination between samples, the presence of the negative control proved to be essential for the interpretation of the data. Sequencing of negative controls revealed the presence of 1,210 sequences (i.e. average value of the three controls) of which 225 matched with C. parvum-IIcA5G3 and C. hominis-IbA10G2. The most represented subtype in the negative control was C. parvum IIdA22G1 reaching 983 sequences. Two hypotheses could explain the high number of this subtype sequences detected in the negative control: (i) almost half of the samples handled in the run were positive for this subtype, and (ii) as previously described, the prevalence of contaminants can be higher in negative controls than in other samples due to the absence of competitive DNA during the sequencing process. Based on this higher value (983 sequences) observed in the negative control, all sequences generated in lower numbers were thus considered as contaminating sequences. Note however that, while mixes C and D were known to contain 1% and 0.1% of C. hominis IbA10G2 respectively, only 777 and 75 sequences respectively were generated by NGS.”

Response: Noted and appreciated. Please see response above for Line 152 and Line 155.

Answer: As requested by the reviewer, we answered to the comments above.

Response: Noted and appreciated.

Answer: We thank the reviewer

Line 155: 1) The decimal symbol is a period in English. Please revise the notation

The notation has been revised as requested by the 2nd reviewer.

Response: Noted and appreciated.

Answer: Ok

Line 155: 2) The total number of NGS sequences of each sample does not match the sum of sequences from all subtypes. Please explain.

As seen in Table 1, the sum of sequences from all subtypes does not match the total number of NGS sequences showed in the last column of the Table 2. The explanation is that in each NGS run, a small number of sequences are not assigned to any sequence in our database, nor to BLAST. They are therefore "Not assigned". To avoid cluttering the table with an extra column, we did not want to put this information in the table. However, we have omitted to put a sentence below the table to explain that there were ‘’Not assigned’’ sequences. We thank the reviewer for allowing us to correct this error. We propose to add the following sentence: « Some sequences generated during the NGS run could not be assigned to any sequence in our database, nor to BLAST. These "Not Assigned" sequences represented an average of 45.2 (between 0 and 81) sequences in each sample of the run (data not shown) » (Line 204). If the editor prefers that the data regarding the "Not assigned sequences" appear in the table, we can add them.

Response: The addition of the description is noted and appreciated. In order to keep the integrity of the data, please add the number of the “not assigned” sequences to Table 2 (Suggestion: number of “not assigned” sequences can be in the parenthesis following the total number in the same column).

Answer: We thank the reviewer for this suggestion and that has been done. The mention of the “not assigned” sequences appears now, as suggested, in the same column as the “total number of sequences” (Line 190). The following sentence appears line 204: “Some sequences generated during the NGS run could not be assigned to any sequence in our database, nor to BLAST (Table 2)”.

Response: Revision noted and appreciated.

Answer: We thank the reviewer

Line 164: Please specify the location of the data in Table S3 and explain the discrepancy of the number between the context and Table S3 for sample 2249.

The location of the data has been specified (Line 202), and the discrepancy of the values has been corrected at line 201. We apologize for this typing error that had not been seen during our proofreading.

Response: Noted and appreciated.

Answer: Ok

Line 228: Other than the definition of other studies, what is your set up criteria for an interpretation threshold in this study and, how would you differentiate the minorities of C. subtypes from sequencing errors?

Based on the observation of our sequencing results, we proposed to set an interpretation threshold based on the negative control results. Indeed, considering all the sequences generated during the sequencing reaction, with the risk of identifying subtypes as true whereas they are not, is not acceptable. Furthermore, setting an arbitrary threshold before the run, as described in the scientific literature, does not seem acceptable either. Thus, using the highest number of sequences obtained in the negative control (983 sequences in the case of this run) which depicts the highest level of contamination, we decided that any sequences with a number lower than 983 would be considered as contaminating sequences. However, as underlined in the “Discussion” section (Line 284), interpretation of the data based on the presence of contaminating sequences in the negative controls can lead to loss of sensitivity of the technique, which can make the presence of minor variants difficult to interpret. In the case of this run, the chosen threshold led to the underestimation of minority variants within three samples (D45, mixes C and D).

To answer the reviewer’s request, we modified the “Discussion” section from line 269 to line 288. These modifications include the following paragraph.

“However, limits regarding the interpretation of the data were observed using NGS methods. Indeed, the presence of unexpected sequences in low number in all the samples, including negative controls, highlighted the need for operator expertise and technical care since most of those unexpected sequences were linked to Cryptosporidium species/subtypes present in other samples handled in the same run. The negative controls thus appeared essential for the interpretation of the data, by setting an interpretation threshold based on the highest number of sequences generated in the negative control during each sequencing run. Indeed, this is not acceptable to consider all the sequences generated during a sequencing run, with the risk of identifying subtypes as true whereas they are not. The use of the negative control to set a detection threshold is however questionable since cross-contaminations can lead to misinterpretations. As example for sample D45 and mixes C and D in which, detection of C. hominis IbA10G2 sequences was expected, but for which sequencing generated fewer sequences than in the negative control. These observations indicate that minority variants present at 1% for C. hominis would have been possibly ignored when setting up an interpretation threshold as described in our study. Finally, these observations highlight the difficulty of setting up a threshold for the interpretation of NGS data and the need for standardization of interpretation thresholds to allow comparison of the results of genotyping studies carried out with the same NGS based tools”.

The reviewer also asked us how we would differentiate sequencing errors from real variants. As mentioned in the “Discussion” section at line 325, the presence of minor subtypes which can not be linked to cross-contamination (as C. parvum IIaA11G2R1) should raise questions about sequencing errors. In the case of this study, only one minor subtype was concerned: C. parvum IIaA11G2R1 found in Grasse outbreak samples. Different arguments suggest that this subtype was not linked to PCR/sequencing error but was a real minor variant.

These arguments are:

(i) MiSeq sequencer has an error rate of 0.9 every 100 bases, which is much lower than other sequencers using different technologies (Stoler N, Nekrutenko A. Sequencing error profiles of Illumina sequencing instruments. NAR Genom Bioinform2021;3(1):lqab019.. doi:10.1093/nargab/lqab019), and which seems insufficient to explain the presence of C. parvum IIaA11G2R1. Indeed, this would require a much higher error rate to generate, for example, a C. parvum IIaA11G2R1 instead of a C. parvum IIaA17G1R1, and above all these errors should only concern the microsatellite region, which seems unlikely.

(ii) PCR slippage has already been described in microsatellite regions. However, it seems unlikely that a PCR slippage generated the same subtype in 6 different samples.

(iii) Finally, the bioinformatics pipeline used is supposed to limit the impact of PCR/sequencing errors on the final data. DADA2 pipeline is described as a “correcting Illumina-sequenced amplicon errors” pipeline, which proceeds to the modeling of the sequencing error, supposed to make it possible to distinguish mutant sequences from erroneous sequences. Thus, as described by Callahan et al. in several mock communities, DADA2 identified more real variants and output fewer spurious sequences than other methods.

To be clearer, the following sentences have been added/modified in the “Discussion” section (Lines 323 to 332) “In parallel, cross-contamination could be excluded for some unexpected sequences thanks to original features shared by these subtypes sequences (i.e. IIaA11G2R1). The presence of minor subtypes which can not be linked to cross-contamination (as C. parvum IIaA11G2R1) should raise questions about sequencing errors. Sequencing errors could be linked to (i) PCR polymerase slippage as suggested elsewhere (20), linked to polymerase elongation mistakes in microsatellites regions, (ii) sequencing errors linked to the MiSeq sequencer, which is known to have a 0.9% error rate (ref). However, that random sequencing errors could generate an identical subtype in 6 samples seems highly unlikely. Additionally, the DADA2 bioinformatics pipeline is supposed to limit the impact of PCR/sequencing errors on the final data. This minority subtype could be due to PCR polymerase slippage as suggested elsewhere (20), linked to polymerase elongation mistakes in microsatellites regions. However, since bioinformatics analyses were strict and other Cryptosporidium sp. positive samples of the same run did not contain mixtures, this suggested real mixtures (20). This minority variant could therefore reflect the end of the clonal epidemic wave, or could be linked Another explanation could be within-host genetic diversity, resulting from mutations or recombination during parasitic life cycle (15).“ Furthermore, the following sentences have been added in the “Materials and Methods” section: “This pipeline is supposed to limit the impact of PCR/sequencing errors on the final data. DADA2 is described as a “correcting Illumina-sequenced amplicon errors” pipeline, which proceeds to the modeling of the sequencing error, supposed to make it possible to distinguish mutant sequences from erroneous sequences. Thus, as described by Callahan et al. in several mock communities, DADA2 identified more real variants and output fewer spurious sequences than other methods.“(Lines 405 to 410).

Response: Noted and appreciated.

Answer: Ok

The suspension of contamination in the negative control seems to be the major hurdle of setting up the threshold in this study. Please see comments above to provide information for confirmation of contamination.

Answer: As requested by the reviewer, we answered to the comments above.

Response: Noted and appreciated.

Answer: We thank the reviewer

Line 258: Please present the NGS data of these 108 DNA samples in the main context instead. This is the strong evidence for comparison of SgS and NGS.

As rightly underlined by the reviewer, it is important to show the comparison between SgS and NGS more explicitly using a table or a figure. However, the table S4 seemed complicated to us to put in the main context, because of its large size. We thus suggest adding Figure 1 (Line 210), which allows comparing the investigation of Grasse outbreak made by SgS or NGS, in terms of variant proportion and mixed infections detection.

Response: Addition of Figure 1 is noted and appreciated. Please also add the statement from the comparison provided in Figure 1.

Answer: As requested, the statement from the comparison has been added line 208, with the following sentences: “Figure 1 compares the investigation of the outbreak carried out by SgS and by NGS, in terms of subtypes proportion and mixed infections detection. By investigating more samples, NGS made it possible to reflect the true proportions of each variant within the outbreak, notably by showing that C. parvum IIdA22G1 represented almost 90% of the subtypes found during the outbreak. Furthermore, NGS was able to highlight mixed infections within the outbreak which had been ignored by SgS”.

Response: Noted and appreciated.

Answer: We thank the reviewer

Line 336: Please use English for notation in Tables.

The notation has been revised as requested by the 2nd reviewer.

Response: Noted and appreciated.

Answer: Ok

Line 365: 1) Check and revise the information of citations in English.

We thank the reviewer for this comment. The relevance of the references already present in the original manuscript has been checked, and numbering has been updated as references were added. A total of six references were added in response to the different reviews.

  1. P. Chaud, L. Ramalli, J. Raibaut, C. Ortmans, S. Joubert, F. Chereau, F. Dalle, S. Valot, D. Costa, A. François, Épidémie de cryp-tosporidiose d’origine hydrique dans les Alpes Maritimes–novembre 2019, Médecine et Maladies Infectieuses, Volume 50, Issue 6, Supplement, 2020, Page S167, ISSN 0399-077X doi:10.1016/j.medmal.2020.06.357.
  1. Sulaiman IM, Hira PR, Zhou L, et al. Unique endemicity of cryptosporidiosis in children in Kuwait. J Clin Microbiol. 2005;43(6):2805-2809. doi:10.1128/JCM.43.6.2805-2809.2005
  1. Rotondo JC, Martini F, Maritati M, et al. Advanced Molecular and Immunological Diagnostic Methods to Detect SARS-CoV-2 Infection. Microorganisms. 2022;10(6):1193. doi:10.3390/microorganisms10061193
  1. Davis NM, Proctor DM, Holmes SP, Relman DA, Callahan BJ. Simple statistical identification and removal of contaminant sequences in marker-gene and metagenomics data. Microbiome. 2018;6(1):226. doi:10.1186/s40168-018-0605-2
  1. Ramo A, Quílez J, Del Cacho E, Sánchez-Acedo C. Optimization of a fragment size analysis tool for identification of Cryptosporidium species and Gp60 alleles infecting domestic ruminants. Vet Parasitol. 2014;205(3-4):466-471. doi:10.1016/j.vetpar.2014.08.025
  1. Stoler N, Nekrutenko A. Sequencing error profiles of Illumina sequencing instruments. NAR Genom Bioinform. 2021;3(1):lqab019.. doi:10.1093/nargab/lqab019

Response: Noted and appreciated.

Answer: Ok

Line 365: 2) Revise the format of references.

As suggested, the reference format has been homogenized and adapted to the recommendations of the "instructions for authors" section of Pathogens Journal website.

Response: Noted and appreciated.

Answer: Ok

This manuscript is a resubmission of an earlier submission. The following is a list of the peer review reports and author responses from that submission.